# Deep Equilibrium Approaches to Diffusion Models

**Ashwini Pokle**
Carnegie Mellon University
`apokle@cs.cmu.edu`

**Zhengyang Geng**
Carnegie Mellon University
`zgeng2@cs.cmu.edu`

**Zico Kolter**
Carnegie Mellon University
Bosch Center for AI
`zkolter@cs.cmu.edu`

## Abstract

Diffusion-based generative models are extremely effective in generating high-quality images, with generated samples often surpassing the quality of those produced by other models under several metrics. One distinguishing feature of these models, however, is that they typically require long sampling chains to produce high-fidelity images. This presents a challenge not only from the lenses of sampling time, but also from the inherent difficulty in backpropagating through these chains in order to accomplish tasks such as model inversion, *i.e.,* approximately finding latent states that generate known images. In this paper, we look at diffusion models through a different perspective, that of a (deep) equilibrium (DEQ) fixed point model. Specifically, we extend the recent denoising diffusion implicit model (DDIM) [68], and model the entire sampling chain as a joint, multi-variate fixed point system. This setup provides an elegant unification of diffusion and equilibrium models, and shows benefits in 1) single image sampling, as it replaces the fully-serial typical sampling process with a parallel one; and 2) model inversion, where we can leverage fast gradients in the DEQ setting to much more quickly find the noise that generates a given image. The approach is also orthogonal and thus complementary to other methods used to reduce the sampling time, or improve model inversion. We demonstrate our method's strong performance across several datasets, including CIFAR10, CelebA, and LSUN Bedroom and Churches.[1]

## 1 Introduction

Diffusion models have emerged as a promising class of generative models that can generate high quality images [69, 68, 57], outperforming GANs on perceptual quality metrics [19], and likelihood-based models on density estimation [42]. One of the limitations of these models, however, is the fact that they require a long diffusion chain (many repeated applications of a denoising process), in order to generate high-fidelity samples. Several recent papers have focused on tackling this limitation, *e.g.,* by shortening the length of diffusion process through an alternative parameterization [68, 44], or through progressive distillation of a sampler with large diffusion chain into a smaller one [54, 65]. However, all of these methods still rely on a fundamentally sequential sampling process, imposing challenges on accelerating the sampling and for other applications like differentiating through the entire generation process.

In this paper, we propose an alternative approach that also begins to address such challenges from a different perspective. Specifically, we propose to model the generative process of a specific class of

---

[1] Code is available at `https://github.com/ashwinipokle/deq-ddim`

36th Conference on Neural Information Processing Systems (NeurIPS 2022).

diffusion model, the denoising diffusion implicit model (DDIM) [68], as a deep equilibrium (DEQ) model [6]. Deep equilibrium (DEQ) models are networks that aim to find the fixed point of the underlying system in the forward pass and differentiate implicitly through this fixed point in the backward pass. To apply DEQs to diffusion models, we first formulate this process as an equilibrium system consisting of all $T$ joint sampling steps, and then *simultaneously* solve the fixed point of all the $T$ steps to achieve sampling.

This approach has several benefits: **First**, the DEQ sampling process can be solved in parallel over multiple GPUs by batching the workload. This is particularly beneficial in the case of single image (*i.e.,* batch-size-one) generation, where the serial sampling nature of diffusion models has inevitably made them unable to maximize GPU computation. **Second**, solving for the joint equilibria simultaneously leads to faster overall convergence as we have better estimates of the intermediate states in fewer steps. Specifically, the formulation naturally lends itself to an augmented diffusion chain that each state is generated according to *all* others. **Third**, the DEQ formulation allows us to leverage a faster differentiation through the chain. This enables us to much more effectively solve problems that require us to differentiate through the generative process, useful for tasks such as *model inversion* that seeks to find the noise that leads to a particular instance of an image.

We demonstrate the advantages of this formulation on two applications: single image generation and model inversion. Both single image generation and model inversion are widely applied in real world image manipulation tasks like editing and restoration [77, 63, 35, 2, 49, 55]. On CIFAR-10 [46] and CelebA [52], DEQ achieves up to $2\times$ speedup over the sequential sampling processes of DDIM [68], while maintaining a comparable perceptual quality of images. In the model inversion, the loss converges much faster when trained with DEQs than with sequential sampling. Moreover, optimizing the sequential sampling process can be computationally expensive. Leveraging modern autograd packages is infeasible as they require storing the entire computational graph for all $T$ states. Some recent works like Nie et al. [58] achieve this through use of SDE solvers. In contrast, with DEQs we can use the implicit differentiation of $\mathcal{O}(1)$ memory complexity. Empirically, the initial hidden state recovered by DEQ more accurately regenerates the original image while capturing its finer details.

To summarize, our main contributions are as follows:

- We formulate the generative process of an augmented type of DDIM as a deep equilibrium model that allows the use of black-box solvers to efficiently compute the fixed point and generate images.

- The DEQ formulation parallelizes the sampling process of DDIM, and as a result, it can be run on multiple GPUs instead of a single GPU. This alternate sampling process converges faster compared to the original process.

- We demonstrate the advantages of our formulation on single image generation and model inversion. We find that optimizing the sampling process via DEQs consistently outperforms naive sequential sampling.

- We provide an easy way to extend this DEQ formulation to a more general family of diffusion models with stochastic generative process like DDPM [67, 33].

## 2 Preliminaries

**Diffusion Models**   Denoising diffusion probabilistic models (DDPM) [67, 33] are generative models that can convert the data distribution to a simple distribution, (*e.g.,* a standard Gaussian, $\mathcal{N}(\mathbf{0}, \mathbf{I})$), through a diffusion process. Specifically, given samples from a target distribution $\mathbf{x}_0 \sim q(\mathbf{x}_0)$, the diffusion process is a Markov chain that adds Gaussian noises to the data to generate latent states $\mathbf{x}_1, ..., \mathbf{x}_T$ in the same sample space as $\mathbf{x}_0$. The inference distribution of diffusion process is given by:

$$q(\mathbf{x}_{1:T}|\mathbf{x}_0) = \prod_{t=1}^{T} q(\mathbf{x}_t|\mathbf{x}_{t-1}) \tag{1}$$

To learn the parameters $\theta$ that characterize a distribution $p_\theta(\mathbf{x}_0) = \int p_\theta(\mathbf{x}_{0:T}) d\mathbf{x}_{1:T}$ as an approximation of $q(\mathbf{x}_0)$, a surrogate variational lower bound [67] was proposed to train this model:

$$L = \mathbb{E}_q[-\log p_\theta(\mathbf{x}_0|\mathbf{x}_1) + \sum_t D_{KL}(q(\mathbf{x}_{t-1}|\mathbf{x}_t, \mathbf{x}_0)||p_\theta(\mathbf{x}_{t-1}|\mathbf{x}_t) + D_{KL}(q(\mathbf{x}_T|\mathbf{x}_0)||p(\mathbf{x}_T)] \tag{2}$$

After training, samples can be generated by a reverse Markov chain, *i.e.,* first sampling $\mathbf{x}_T \sim p(\mathbf{x}_T)$, and then repeatedly sampling $\mathbf{x}_{t-1}$ till we reach $\mathbf{x}_0$.

As noted in [67, 68], the length $T$ of a diffusion process is usually large (*e.g.,* $T = 1000$ [33]) as it contributes to a better approximation of Gaussian conditional distributions in the generative process. However, because of the large value of $T$, sampling from diffusion models can be visibly slower compared to other deep generative models like GANs [29].

One feasible acceleration is to rewrite the forward process into a non-Markovian one that leads to a "shorter" and deterministic generative process, *i.e.,* denoising diffusion implicit model [68] (DDIM). DDIM can be trained similarly to DDPM, using the variational lower bound shown in Eq. (2). Essentially, DDIM constructs a nearly non-stochastic scheme that can quickly sample from the learned data distribution without introducing additional noises. Specifically, the scheme to generate a sample $\mathbf{x}_{t-1}$ given $\mathbf{x}_t$ is:

$$\mathbf{x}_{t-1} = \sqrt{\alpha_{t-1}}\left(\frac{\mathbf{x}_t - \sqrt{1-\alpha_t}\boldsymbol{\epsilon}_\theta^{(t)}(\mathbf{x}_t)}{\sqrt{\alpha_t}}\right) + \sqrt{1-\alpha_{t-1} - \sigma_t^2} \cdot \boldsymbol{\epsilon}_\theta^{(t)}(\mathbf{x}_t) + \sigma_t\boldsymbol{\epsilon}_t \tag{3}$$

where $\alpha_1, ..., \alpha_T \in (0, 1]$, $\boldsymbol{\epsilon}_t \sim \mathcal{N}(\mathbf{0}, \mathbf{I})$, and $\boldsymbol{\epsilon}_\theta^{(t)}(\mathbf{x}_t)$ is an estimator trained to predict the noise given a noisy state $\mathbf{x}_t$. Different values of $\sigma_t$ define different generative processes. For a variance schedule $\beta_1 \ldots \beta_T$, we use the notation $\alpha_t = \prod_{s=1}^t(1 - \beta_s)$. When $\sigma_t = \sqrt{(1-\alpha_{t-1})/(1-\alpha_t)}\sqrt{1 - \alpha_t/\alpha_{t-1}}$ for all $t$, the generative process represents a DDPM. Setting $\sigma_t = 0$ for all $t$ gives rise to a DDIM, which results in a deterministic generating process except the initial sampling $\mathbf{x}_T \sim p(\mathbf{x}_T)$.

**Deep Equilibrium Models**   Deep equilibrium models are a recently-proposed class of deep networks that, in their forward pass, seek to find a fixed point of a single layer applied repeatedly to a hidden state. Specifically, consider a deep feedforward model with $L$ layers:

$$\mathbf{z}^{[i+1]} = f_\theta^{[i]}\left(\mathbf{z}^{[i]}; \mathbf{x}\right) \quad \text{for } i = 0, ..., L-1 \tag{4}$$

where $\mathbf{x}$ is the input injection, $\mathbf{z}^{[i]}$ is the hidden state of $i^{th}$ layer, and $f_\theta^{[i]}$ is a layer that defines the feature transformation. Assuming the above model is weight-tied, *i.e.,* $f_\theta^{[i]} = f_\theta, \forall i$, then in the limit of infinite depth, the output $\mathbf{z}^{[i]}$ of this network converges to a fixed point $\mathbf{z}^*$.

$$\lim_{i \to \infty} f_\theta\left(\mathbf{z}^{[i]}; \mathbf{x}\right) = \mathbf{z}^* \tag{5}$$

Inspired from the neural convergence phenomenon, Deep equilibrium (DEQ) models [6] are proposed to directly compute this fixed point $\mathbf{z}^*$ as the output, *i.e.,*

$$f_\theta(\mathbf{z}^*; \mathbf{x}) = \mathbf{z}^* \tag{6}$$

The equilibrium state $\mathbf{z}^*$ can be solved by black-box solvers like Broyden's method [13], or Anderson acceleration [5]. To train this fixed-point system, Bai et al. [6] leverage implicit differentiation to directly backpropagate through the equilibrium state $\mathbf{z}^*$ using $\mathcal{O}(1)$ memory complexity. DEQ is known as a principled framework for characterizing convergence and energy minimization in deep learning. We leave a detailed discussion in Sec. 6.

## 3   A Deep Equilibrium Approach to DDIMs

In this section, we present the main modeling contribution of the paper, a formulation of diffusion processes under the DEQ framework. Although diffusion models may seem to be a natural fit for DEQ modeling (after all, we typically *do not* care about intermediate states in the denoising chain, but only the final clean image), there are several reasons why setting up the diffusion chain "naively" as a DEQ (*i.e.,* making $f_\theta$ be a single sampling step) does not ultimately lead to a functional algorithm. Most fundamentally, the diffusion process is not time-invariant (*i.e.,* not "weight-tied" in the DEQ sense), and the final generated image is practically-speaking independent of the noise used to generate it (*i.e.,* not truly based upon "input injection" either).

Thus, at a high level, our approach to building a DEQ version of the DDIM involves representing *all* the states $\mathbf{x}_{0:T}$ *simultaneously* within the DEQ state. The advantage of this approach is that 1)

we can exactly capture the typical diffusion inference chain; and 2) we can create a *more expressive* reverse process where the state $\mathbf{x}_t$ is updated based upon *all* previous states $\mathbf{x}_{t+1:T}$, improving the inference process; 3) we can execute all steps of the inference chain *in parallel* rather than solely in sequence as is typically required in diffusion models; and 4) we can use common DEQ acceleration methods, such as the Anderson solver [5] to find the fixed point, which makes the sampling process converge faster. A downside of this formulation is that we need to store all DEQ states simultaneously (*i.e., only* the images, not the intermediate network states).

## 3.1  A DEQ formulation of DDIMs (DEQ-DDIM)

The generative process of DDIM is given by:

$$\mathbf{x}_{t-1} = \sqrt{\alpha_{t-1}}\left(\frac{\mathbf{x}_t - \sqrt{1-\alpha_t}\boldsymbol{\epsilon}_\theta^{(t)}(\mathbf{x}_t)}{\sqrt{\alpha_t}}\right) + \sqrt{1-\alpha_{t-1}}\cdot\boldsymbol{\epsilon}_\theta^{(t)}(\mathbf{x}_t), \quad t = [1,\ldots,T] \qquad (7)$$

This process also lets us generate a sample using a subset of latent states $\{\mathbf{x}_{\tau_1},\ldots,\mathbf{x}_{\tau_S}\}$, where $\{\tau_1,\ldots,\tau_S\} \subseteq T$. While this helps in accelerating the overall generative process, there is a tradeoff between sampling quality and computational efficiency. As noted in Song et al. [68], larger $T$ values lead to lower FID scores of the generated images but need more compute time; smaller $T$ are faster to sample from, but the resulting images have worse FID scores.

Reformulating this sampling process as a DEQ addresses multiple concerns raised above. We can define a DEQ, with a sequence of latent states $\mathbf{x}_{1:T}$ as its internal state, that *simultaneously* solves for the equilibrium points at all the timesteps. The global convergence of this process is upper bounded by $T$ steps, by definition. To derive the DEQ formulation of the generative process, first we rearrange the terms in Eq. (7):

$$\mathbf{x}_{t-1} = \sqrt{\frac{\alpha_{t-1}}{\alpha_t}}\mathbf{x}_t + \left(\sqrt{1-\alpha_{t-1}} - \sqrt{\frac{\alpha_{t-1}(1-\alpha_t)}{\alpha_t}}\right)\epsilon_\theta^{(t)}(\mathbf{x}_t) \qquad (8)$$

Let $c_1^{(t)} = \sqrt{1-\alpha_{t-1}} - \sqrt{\frac{\alpha_{t-1}(1-\alpha_t)}{\alpha_t}}$. Then we can write

$$\mathbf{x}_{t-1} = \sqrt{\frac{\alpha_{t-1}}{\alpha_t}}\mathbf{x}_t + c_1^{(t)}\epsilon_\theta^{(t)}(\mathbf{x}_t) \qquad (9)$$

By induction, we can rewrite the above equation as:

$$\mathbf{x}_{T-k} = \sqrt{\frac{\alpha_{T-k}}{\alpha_T}}\mathbf{x}_T + \sum_{t=T-k}^{T-1}\sqrt{\frac{\alpha_{T-k}}{\alpha_t}}c_1^{(t+1)}\epsilon_\theta^{(t+1)}(\mathbf{x}_{t+1}), \quad k \in [0,..,T] \qquad (10)$$

This defines a "fully-upper-triangular" inference process, where the update of $\mathbf{x}_t$ depends on the noise prediction network $\epsilon_\theta$ applied to *all* subsequent states $\mathbf{x}_{t+1:T}$; in contrast to the traditional diffusion process, which updates $\mathbf{x}_t$ based only on $\mathbf{x}_{t+1}$. Specifically, let $h(\cdot)$ represent the function that performs the operations in the equations (10) for a latent $\mathbf{x}_t$ at timestep $t$, and let $\tilde{h}(\cdot)$ represent the function that performs the same set of operations across all the timesteps simultaneously. We can write the above set of equations as a fixed point system:

$$\begin{bmatrix}\mathbf{x}_{T-1}\\\mathbf{x}_{T-2}\\\vdots\\\mathbf{x}_0\end{bmatrix} = \begin{bmatrix}h(\mathbf{x}_T)\\h(\mathbf{x}_{T-1:T})\\\vdots\\h(\mathbf{x}_{1:T})\end{bmatrix}$$

or,

$$\mathbf{x}_{0:T-1} = \tilde{h}(\mathbf{x}_{0:T-1};\mathbf{x}_T) \qquad (11)$$

The above system of equations represent a DEQ with $\mathbf{x}_T \sim \mathcal{N}(\mathbf{0},\mathbf{I})$ as input injection. We can simultaneously solve for the roots of this system of equations through black-box solvers like Anderson acceleration [5]. Let $g(\mathbf{x}_{0:T-1};\mathbf{x}_T) = \tilde{h}(\mathbf{x}_{0:T-1};\mathbf{x}_T) - \mathbf{x}_{0:T-1}$, then we have

$$\mathbf{x}_{0:T}^* = \text{RootSolver}(g(\mathbf{x}_{0:T-1};\mathbf{x}_T)) \qquad (12)$$

This DEQ formulation has multiple benefits. Solving for all the equilibria simultaneously leads to a better estimation of the intermediate latent states $\mathbf{x}_t$ in a fewer number of steps (*i.e.,* $\leq t$ steps for $\mathbf{x}_t$). This leads to faster convergence of the sampling process as the final sample $\mathbf{x}_0$, which is dependent on the latent states of all the previous time steps, has a better estimate of these intermediate latent states. Note that by the same reasoning, the intermediate latent states $\mathbf{x}_t$ converge faster too. Thus, we can get images with perceptual quality comparable to DDIM in a significantly fewer number of steps. Of course, we also note that the computational requirements of each individual step has significantly increased, but this is at least largely offset by the fact that the steps can be executed as mini-batched in parallel over each state. Empirically, in fact, we often notice significant *speedup* using this approach on tasks like single image generation.

This DEQ formulation of DDIM can be extended to the stochastic generative processes of DDIM with $\eta > 0$, including that of DDPM (referred to as DEQ-sDDIM). The key idea is to sample noises for all the time steps along the sampling chain and treat this noise as an input injection to DEQ, in addition to $\mathbf{x}_T$.

$$\mathbf{x}_{0:T}^* = \text{RootSolver}(g(\mathbf{x}_{0:T-1}; \mathbf{x}_T, \boldsymbol{\epsilon}_{1:T})) \tag{13}$$

where RootSolver($\cdot$) is any black-box fixed point solver, and $\boldsymbol{\epsilon}_{1:T} \sim \mathcal{N}(\mathbf{0}, \mathbf{I})$ represents the input injected noises. We discuss this formulation in more detail in Appendix D.

## 4 Efficient Inversion of DDIM

One of the primary strengths of DEQs is their constant memory consumption, for both forward pass and backward pass, regardless of their 'effective depth'. This leads to an interesting application of DEQs in inverting DDIMs that fully leverages this advantage along with the other benefits discussed in the previous section.

| **Algorithm 1** A naive algorithm to invert DDIM | **Algorithm 2** Inverting DDIM with DEQ |
|---|---|
| **Input:** A target image $\mathbf{x}_0 \sim \mathcal{D}$, $\boldsymbol{\epsilon}_\theta(\mathbf{x}_t, t)$ a trained denoising diffusion model, $N$ the total number of epochs | **Input:** A target image $\mathbf{x}_0 \sim \mathcal{D}$, $\boldsymbol{\epsilon}_\theta(\mathbf{x}_t, t)$ a trained denoising diffusion model, $N$ the total number of epochs |
| $\triangleright f$ denotes the sampling process in Eq (7) | $\triangleright g$ is the function in Eq. (12) |
| **Initialize** $\hat{\mathbf{x}}_T \sim \mathcal{N}(\mathbf{0}, \mathbf{I})$ | **Initialize** $\hat{\mathbf{x}}_{0:T} \sim \mathcal{N}(\mathbf{0}, \mathbf{I})$ |
| **for** epochs from 1 to $N$ **do** | **for** epochs from 1 to $N$ **do** |
|     **for** $t = T, ..., 1$ **do** |     $\triangleright$ Disable gradient computation |
|         Sample $\hat{\mathbf{x}}_{t-1} = f(\hat{\mathbf{x}}_t; \epsilon_\theta(\hat{\mathbf{x}}_t, t))$ |     $\mathbf{x}_{0:T}^* = \text{RootSolver}(g(\mathbf{x}_{0:T-1}); \mathbf{x}_T)$ |
|     **end for** |     $\triangleright$ Enable gradient computation |
|     Take a gradient descent step on |     Compute Loss $\mathcal{L}(\mathbf{x}_0, \mathbf{x}_0^*)$ |
|     $\nabla_{\hat{\mathbf{x}}_T} \|\hat{\mathbf{x}}_0 - \mathbf{x}_0\|_F^2$ |     Use the 1-step grad to compute $\partial \mathcal{L} / \partial \mathbf{x}_T$ |
| |     Take a gradient descent step using above |
| **end for** | **end for** |
| **Output:** $\hat{\mathbf{x}}_T$ | **Output:** $\mathbf{x}_T^*$ |

### 4.1 Problem Setup

Given an arbitrary image $\mathbf{x}_0 \sim \mathcal{D}$, and a denoising diffusion model $\boldsymbol{\epsilon}_\theta(\mathbf{x}_t, t)$ trained on a dataset $\mathcal{D}$, model inversion seeks to determine the latent $\hat{\mathbf{x}}_T \sim \mathcal{N}(\mathbf{0}, \mathbf{I})$ that can generate an image $\hat{\mathbf{x}}_0$ identical to the original image $\mathbf{x}_0$ through the generative process for DDIM described in Eq. (7). For an input image $\mathbf{x}_0$, and a generated image $\hat{\mathbf{x}}_0$, this task needs to minimize the squared-Frobenius distance between these images:

$$\mathcal{L}(\mathbf{x}_0, \hat{\mathbf{x}}_0) = \|\mathbf{x}_0 - \hat{\mathbf{x}}_0\|_F^2 \tag{14}$$

### 4.2 Inverting DDIM: The Naive Approach

A relatively straightforward way to invert DDIM is to randomly sample $\mathbf{x}_T \sim \mathcal{N}(\mathbf{0}, \mathbf{I})$, and update it via gradient descent by first estimating $\mathbf{x}_0$ using the generative process in Eq. (7) and backpropagating through this process after computing the loss objective in (14). The overall process has been summarized in Algorithm 1. This process has a large computational overhead. Every training epoch

requires a sequential sampling for all $T$ timesteps. Optimizing through this generative process would require the creation of a large computational graph for storing relevant intermediate variables necessary for the backward pass. Sequential sampling further slows down the entire process.

### 4.3 Efficient Inversion of DDIM with DEQs

Alternatively, we can use the DEQ formulation to develop a much more efficient inversion method. We provide a high-level overview of this approach in Algorithm 2. We can apply implicit function theorem (IFT) to the fixed point, *i.e.,* (12) to compute gradients of the loss $\mathcal{L}(\mathbf{x}_0, \mathbf{x}_0^*)$ in (14) *w.r.t.* $(\cdot)$:

$$\frac{\partial \mathcal{L}}{\partial (\cdot)} = -\frac{\partial \mathcal{L}}{\partial \mathbf{x}_{0:T}^*} \left( J_{g_\theta}^{-1} \big|_{\mathbf{x}_{0:T}^*} \right) \frac{\partial \tilde{h}(\mathbf{x}_{0:T-1}^*; \mathbf{x}_T)}{\partial (\cdot)} \tag{15}$$

where $(\cdot)$ could be any of the latent states $\mathbf{x}_1, ..., \mathbf{x}_T$, and $J_{g_\theta}^{-1}\big|_{\mathbf{x}_{0:T}^*}$ is the inverse Jacobian of $g(\mathbf{x}_{0:T-1}; \mathbf{x}_T)$ evaluated at $x_{0:T}^*$. Refer to [6] for a detailed proof. Computing the inverse of Jacobian matrix can become computationally intractable, especially when the latent states $\mathbf{x}_t$ are high dimensional. Further, prior works [6, 8, 28] have reported growing instability of DEQs during training due to the ill-conditioning of Jacobian. Recent works [27, 26, 28, 9] suggest that we do not need an exact gradient to train DEQs. We can instead use an approximation to Eq. (15), *i.e.,*

$$\frac{\partial \mathcal{L}}{\partial (\cdot)} = -\frac{\partial \mathcal{L}}{\partial \mathbf{x}_{0:T}^*} \mathbf{M} \frac{\partial \tilde{h}(\mathbf{x}_{0:T-1}^*; \mathbf{x}_T)}{\partial (\cdot)} \tag{16}$$

where $\mathbf{M}$ is an approximation of $J_{g_\theta}^{-1}\big|_{\mathbf{x}_{0:T}^*}$. For example, [27, 26, 28] show that setting $\mathbf{M} = \mathbf{I}$, *i.e.,* 1-step gradient, works well. In this work, we follow Geng et al. [28] to further add a damping factor to the 1-step gradient. The forward pass is given by:

$$\mathbf{x}_{0:T}^* = \text{RootSolver}(g(\mathbf{x}_{0:T-1}); \mathbf{x}_T) \tag{17}$$

$$\mathbf{x}_{0:T}^* = \tau \cdot \tilde{h}(\mathbf{x}_{0:T-1}^*; \mathbf{x}_T^*) + (1 - \tau) \cdot \mathbf{x}_{0:T}^* \tag{18}$$

The gradients for the backward pass can be computed through standard autograd packages. We provide the PyTorch-style pseudocode of our approach in the Appendix B. Using inexact gradients for the backward pass has several benefits: 1) It remarkably improves the training stability of DEQs; 2) Our backward pass consists of a single step and is ultra-cheap to compute. It reduces the total training time by a significant amount. It is easy to extend the strategy used in Algorithm 2 and use DEQs to invert DDIMs with stochastic generative process (referred to as DEQ-sDDIM). We provide the key steps of this approach in Algorithm 4.

## 5 Experiments

We consider four datasets that have images of different resolutions for our experiments: CIFAR10 (32×32) [46], CelebA (64×64) [52], LSUN Bedroom (256×256) and LSUN Outdoor Church (256×256) [76]. For all the experiments, we use Anderson acceleration as the default fixed point solver. We use the pretrained denoising diffusion models from Ho et al. [33] for CIFAR10, LSUN Bedroom, and LSUN Outdoor Church, and from Song et al. [68] for CelebA. While training DEQs for model inversion, we use the 1-step gradient Eq. (18) to compute the backward pass. The damping factor $\tau$ for 1-step gradient is set to $0.1$. All the experiments have been performed on NVIDIA RTX A6000 GPUs. We provide additional experimental details in the Appendix A. While the primary focus in this section will be on the DDIM with a deterministic generative process *i.e.,* $\eta = 0$, we also include a few key results on stochastic version of DDIM (DEQ-sDDIM) here. More extensive experiments can be found in Appendix D.

### 5.1 Convergence of DEQ-DDIM

We verify that DEQ converges to a fixed point by plotting the values of $\|\tilde{h}(\mathbf{x}_{0:T}) - \mathbf{x}_{0:T}\|_2$ over Anderson solver steps. As seen in Figure 1, DEQ converges to a fixed point for generative processes of different lengths. It is easier to reach simultaneous equilibria on smaller sequence lengths than larger sequence lengths. However, this does not affect the quality of images generated. We visualize

the latent states of DEQ in Figure 2. Our experiments demonstrate that DEQ is able to generate high-quality images in as few as 15 Anderson solver steps on diffusion chains that were trained on a much larger number of steps $T$. One might note that DEQs converge to a limit cycle for diffusion processes with larger sequence lengths. This is not a limitation as we only want the latent states at the last few timesteps to converge well, which happens in practice as demonstrated in Fig. 2. Further, these residuals can be driven down by using more powerful solvers like quasi-Newton methods, *e.g.,* Broyden's method.

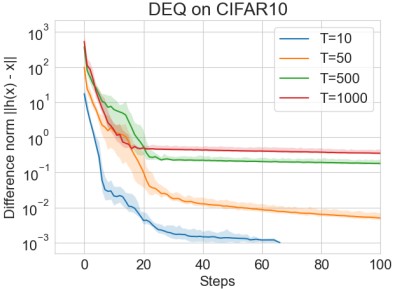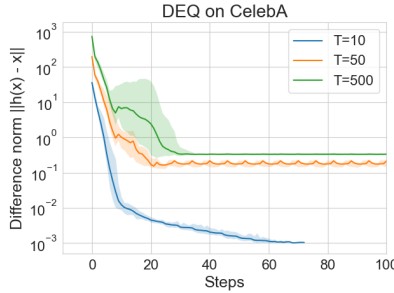

Figure 1: DEQ-DDIM finds an equilibrium point. We plot the absolute fixed-point convergence $\|\tilde{h}(\mathbf{x}_{0:T}) - \mathbf{x}_{0:T}\|_2$ during a forward pass of DEQ for CIFAR-10 (left) and CelebA (right) for different number of steps $T$. The shaded region indicates the maximum and minimum value encountered during any of the 25 runs.

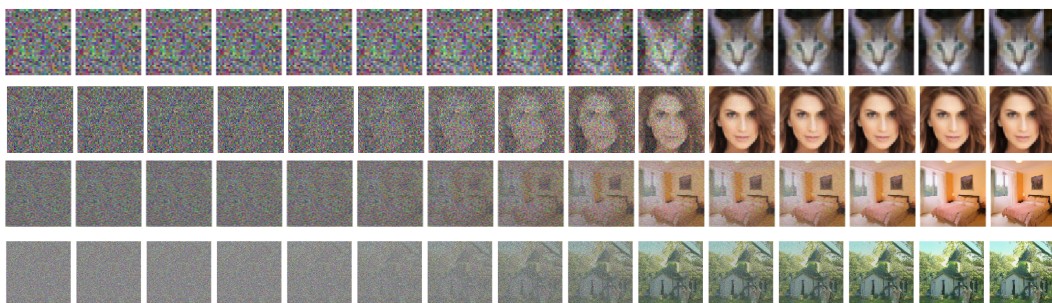

Figure 2: Visualization of intermediate latents $\mathbf{x}_t$ of DEQ-DDIM after 15 forward steps with Anderson solver for CIFAR-10 (first row, $T = 500$), CelebA (second row, $T = 500$), LSUN Bedroom (third row, $T = 50$, and LSUN Outdoor Church (fourth row, $T = 50$). For $T = 500$, we visualize every $50^{th}$ latent, and for $T = 50$, we visualize every $5^{th}$ latent. In addition, we also visualize $\mathbf{x}_{0:4}$ in the last 5 columns.

## 5.2 Sample quality of images generated with DEQ

We verify that DEQs can generate images of comparable quality to DDIM by reporting Fréchet Inception Distance (FID) [32] in Table 1. For the forward pass of DEQ, we run Anderson solver for a maximum of 15 steps for each image. We report FID scores on 50,000 images, and average time to generate an image (including GPU time) on 500 images. We note significant gains in wall-clock time on single-shot image generation with DEQs on images with smaller resolutions. Specifically, DEQs can generate images almost $2\times$ faster than the sequential sampling of DDIM on CIFAR-10 ($32\times32$) and CelebA ($64\times64$). We note that these gains vanish on sequences of shorter lengths and images with larger resolutions as seen in case of LSUN Bedrooms, and Outdoor Churches ($256\times256$). This is because the number of fixed point solver iterations needed for convergence becomes comparable to the length of diffusion chain for small values of $T$. Thus, lightweight updates performed on short diffusion chains for sequential sampling are faster compared to compute heavy updates in DEQs.

We also report FID scores on DEQ-sDDIM for CIFAR10 in Table 2. We run Anderson solver for a maximum of 50 steps for each image. We observe that while DEQ-sDDIM is slower than DDIM,

it always generates images with comparable or better FID scores. For higher levels of stochasticity *i.e.,* for larger valued of $\eta$, DEQ-sDDIM need more Anderson solver iterations to converge to a fixed point, which increases image generation wall-clock time. We include additional results in Appendix D.2. Finally, we also find that on full-batch inference with larger batches, sequential sampling might outperform DEQs, as DEQs would have larger memory requirements in this case, *i.e.,* processing smaller batches of size $B$ might be faster than processing larger batches of size $BT$.

| Dataset | T | DDPM | | DDIM | | DEQ-DDIM | |
|---|---|---|---|---|---|---|---|
| | | FID | Time | FID | Time | FID | Time |
| CIFAR10 | 1000 | **3.17** | 24.45s | 4.07 | 20.16s | 3.79 | **2.91s** |
| CelebA | 500 | 5.32 | 14.95s | 3.66 | 10.31s | **2.92** | **5.12s** |
| LSUN Bedroom | 25 | 184.05 | 1.72s | 8.76 | **1.19s** | **8.73** | 3.82s |
| LSUN Church | 25 | 122.18 | 1.77s | **13.44** | **1.68s** | 13.55 | 3.99s |

Table 1: FID scores and time for single image generation for DDPM, DDIM and DEQ-DDIM.

| $\eta$ | $T$ | FID Scores | | Time (in seconds) | |
|---|---|---|---|---|---|
| | | DDIM | DEQ-sDDIM | DDIM | DEQ-sDDIM |
| 0.2 | 20 | 7.19 | **6.99** | **0.33** | 0.51 |
| 0.5 | 20 | 8.35 | **8.22** | **0.35** | 0.51 |
| 1 | 20 | 18.37 | **17.72** | **0.34** | 0.93 |
| 0.2 | 50 | 4.69 | **4.44** | **0.88** | 0.88 |
| 0.5 | 50 | 5.26 | **4.99** | **0.83** | 1.00 |
| 1 | 50 | 8.02 | **7.85** | **0.83** | 1.58 |

Table 2: FID scores for single image generation for DDIM and DEQ-sDDIM on CIFAR10. Note that DDPM [33] with a larger variance achieves FID scores of $133.37^*$ and $32.72^*$ respectively for $T = 20$ and $T = 50$, where $*$ indicates numbers reported from Song et al. [68]
.

## 5.3 Model Inversion of DDIM with DEQs

We report the minimum values of squared Frobenius norm between the recovered and target images averaged from 100 different runs in Table 3. We report results for DEQ with $\eta = 0$ (*i.e.,* DEQ-DDIM) in this table, and additional results for $\eta > 0$ (*i.e.,* DEQ-sDDIM) are reported in Figure 17. DEQ outperforms the baseline method on all the datasets by a significant margin. We also plot the training loss curves of DEQ-DDIM and the baseline in Figure 3. We observe that DEQ-DDIM converges faster and has much lower loss values than the baseline method induced by DDIM. We also visualize the images generated with the recovered latent states for DEQ-DDIM in Figure 4 and with DEQ-sDDIM in Figure 5. It is worth noting that images generated with DEQ capture more vivid details of the original images, like textures of foliage, crevices, and other finer details than the baseline. We include additional results of model inversion with DEQ-sDDIM on different datasets in Appendix D.3.

| Dataset | T | Baseline | | DEQ-DDIM | |
|---|---|---|---|---|---|
| | | Min loss ↓ | Avg Time (mins) ↓ | Min loss ↓ | Avg Time (mins) ↓ |
| CIFAR10 | 100 | $15.74 \pm 8.7$ | $49.07 \pm 1.76$ | $\mathbf{0.76 \pm 0.35}$ | $12.99 \pm 0.97$ |
| CIFAR10 | 10 | $2.59 \pm 3.67$ | $14.36 \pm 0.26$ | $\mathbf{0.68 \pm 0.32}$ | $2.54 \pm 0.41$ |
| CelebA | 20 | $14.13 \pm 5.04$ | $30.09 \pm 0.57$ | $\mathbf{1.03 \pm 0.37}$ | $28.09 \pm 1.76$ |
| Bedroom | 10 | $1114.49 \pm 795.86$ | $26.41 \pm 0.17$ | $\mathbf{36.37 \pm 22.86}$ | $33.7 \pm 1.05$ |
| Church | 10 | $1674.68 \pm 1432.54$ | $29.7 \pm 0.75$ | $\mathbf{47.94 \pm 24.78}$ | $33.54 \pm 3.02$ |

Table 3: Comparison of minimum loss and average time required to generate an image. All the results have been reported on 100 images. See Appendix A for detailed training settings.

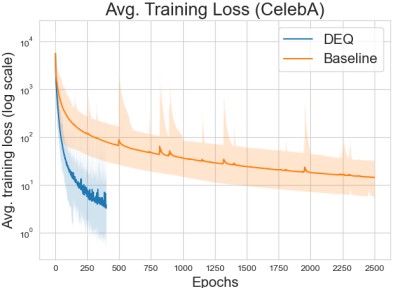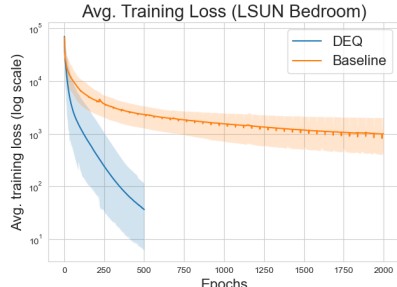

Figure 3: Training loss for CelebA and LSUN Bedroom over epochs. DEQ converges in fewer epochs, and achieves lower values of loss compared to the baseline. The shaded region indicates the maximum and minimum value of loss encountered during any of the 100 runs.

## 6 Related Work

**Implicit Deep Learning** Implicit deep learning is an emerging field that introduces structured methods to construct modern neural networks. Different from prior explicit counterparts defined by hierarchy or layer stacking, implicit models take advantage of dynamical systems [43, 23, 3], *e.g.,* optimization [4, 72, 20, 27, 21], differential equation [16, 22, 70, 30], or fixed-point system [6, 7, 31]. For instance, Neural ODE [16] describes a continuous time-dependent system, while Deep Equilibrium (DEQ) model [6], which is actually path-independent, is a new type of implicit models that outputs the equilibrium states of the underlying system, *e.g.,* $\mathbf{z}^*$ from $\mathbf{z}^* = f_\theta(\mathbf{z}^*, \mathbf{x})$ given the input $\mathbf{x}$. This fixed-point system can be solved by black-box solvers [5, 13], and further accelerated by the neural solver [10] in the inference. An active topic is the stability [8, 28, 9] of such a system as it will gradually deteriorate during training, albeit strong performances [16, 6, 8]. DEQ has achieved SOTA results on a wide-range of tasks like language modeling [6], semantic segmentation [7], graph modeling [31, 51, 59, 15], object detection [73], optical flow estimation [9], robustness [74, 48], and generative models like normalizing flow [53], with theoretical guarantees [75, 38, 25, 50].

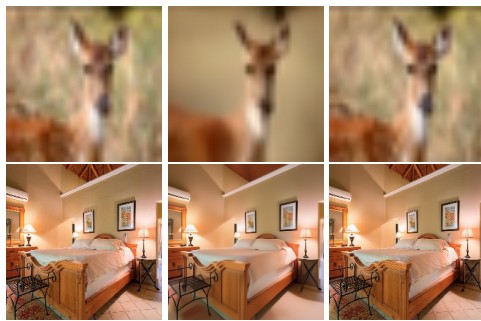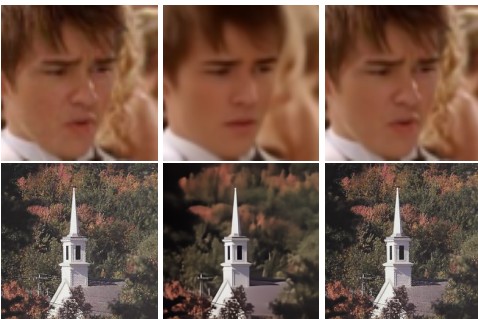

Figure 4: Model inversion on CIFAR10, CelebA, LSUN Bedrooms and Churches, respectively. Each triplet has the original image (left), DDIM's inversion (middle), and DEQ-DDIM's inversion (right).

**Diffusion Models** Diffusion models [67, 33, 68], or score-based generative models [69, 71], are newly developed generative models that utilize an iterative denoising process to progressively sample from a learned data distribution, which actually is the reverse of a forward diffusion process. They have demonstrated impressive fidelity for text-conditioned image generation [62] and outperformed state-of-the-art GANs on ImageNet [19]. Despite the superior practical results, diffusion models suffer from a plodding sampling speed, *e.g.,* over hours to generate 50k CIFAR-sized images [68]. To accelerate the sampling from diffusion models, researchers propose to skip a part of the sampling steps by reframing the reverse chain [68, 45, 44], or distill the trained diffusion model into a faster one [54, 65]. Plus, the forward and backward processes in diffusion models can be formulated as stochastic differential equations [71], bridging diffusion models with Neural ODEs [16] in implicit deep learning. However, the community still lacks insights into the connection between DEQ and diffusion models, where we build our work to investigate this.

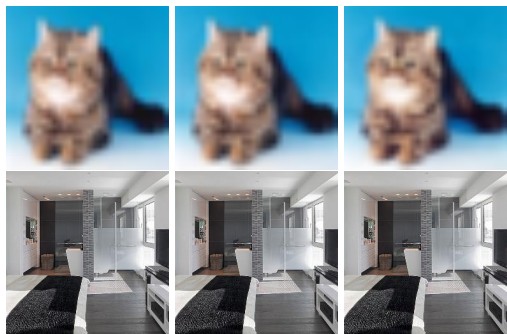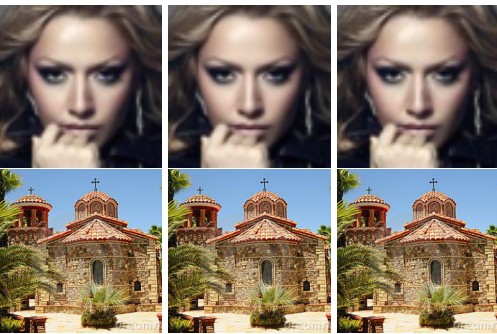

Figure 5: Model inversion of DEQ-sDDIM on CIFAR10, CelebA, LSUN Bedrooms and Churches, respectively. Each triplet displays the original image (left), and images obtained through inversion with DEQ-sDDIM for $\eta = 0.5$ (middle), and $\eta = 1$ (right).

**Model inversion**    Model inversion gives insights into the latent space of a generative model, as an inability of a generative model to correctly reconstruct an image from its latent code is indicative of its inability to model all the attributes of image correctly. Further, the ability to manipulate the latent codes to edit high-level attributes of images finds applications in many tasks like semantic image manipulation [77, 2], super resolution [14, 47], in-painting [18], compressed sensing [12], *etc.* For generative models like GANs [29], inversion is non-trivial and requires alternate methods like learning the mapping from an image to the latent code [11, 61, 78], and optimizing the latent code through optimizers, *e.g.,* both gradient-based [1] and gradient-free [34]. For diffusion models like DDPM [33], the generative process is stochastic, which can make model inversion very challenging. Many existing works based on diffusion models [55, 18, 71, 36, 40] edit images or solve inverse problems without requiring full model inversion. Instead, they do so by utilizing existing understanding of diffusion models as presented in some recent works [71, 37, 39]. Diffusion models have been widely applied to conditional image generation [17, 18, 57, 64, 36, 66, 40, 56]. Chung et al. [18] propose a method to reduce the number of steps in reverse conditional diffusion process through better initialization, based on the idea of contraction theory of stochastic differential equations. Our proposed method is orthogonal to this work; we explicitly model DDIM as a joint, multi-variate fixed point system and leverage black-box root solvers to solve for the fixed point and also allow for efficient differentiation.

## 7    Conclusion

We propose an approach to elegantly unify diffusion models and deep equilibrium (DEQ) models. We model the entire sampling chain of the denoising diffusion implicit model (DDIM) as a joint, multi-variate (deep) equilibrium model. This setup replaces the traditional sequential sampling process with a parallel one, thereby enabling us to enjoy speedup obtained from multiple GPUs. Further, we can leverage inexact gradients to optimize the entire sampling chain quickly, which results in significant gains in model inversion. We demonstrate the benefits of this approach on 1) single-shot image generation, where we were able to obtain FID scores on par with or slightly better than those of DDIM; and 2) model inversion, where we achieved much faster convergence. We also propose an easy way to extend DEQ formulation for deterministic DDIM to its stochastic variants. It is possible to further speedup the sampling process by training a DEQ model to predict the noise at a particular timestep of the diffusion chain. We can jointly optimize the noise prediction network, and the latent variables of the diffusion chain, which we leave as future work.

## 8    Acknowledgements

Ashwini Pokle is supported by a grant from the Bosch Center for Artificial Intelligence.

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
