# A   Experimental Details

In this section, we present detailed settings for all the experiments in Section 5.

**Architecture**   We use exactly the same U-Net [24] architecture for $\epsilon_\theta(\mathbf{x}_t, t)$ as the one previously used by Ho et al. [33], Song et al. [68]. We use pretrained models from Ho et al. [33] for CIFAR10, LSUN Bedrooms and Outdoor Churches, and from Song et al. [68] for CelebA.

**General setting**   We follow the linear selection procedure to select a subsequence of timesteps $\tau_S \subset T$ for all the datasets except CIFAR10, *i.e.,* we select timesteps such that $\tau_i = \lfloor ci \rfloor$ for some $c$. For CIFAR10, we select timesteps such that $\tau_i = \lfloor ci^2 \rfloor$ for some $c$. The constant $c$ is selected so that $\tau_{-1}$ is close to $T$. We use Anderson acceleration [5, 43] as our fixed-point solver for all the experiments. We set the exiting equilibrium error of solver to 1e-3 and set the history length to 5. We allow a maximum of 15 solver forward steps in all the experiments with DEQ-DDIM, and use a maximum of 50 solver forward steps for DEQ-sDDIM. Finally, we use PyTorch's inbuilt DataParallel module to handle parallelization. We use upto 4 NVIDIA Quadro RTX 8000 or RTX A6000 GPUs for all our experiments.

**Training details for model inversion**   We implement and test the code in PyTorch version 1.11.0. We use the Adam [41] optimizer with a learning rate of 0.01. We train DEQs for 400 epochs on CIFAR10 and CelebA, and for 500 epochs on LSUN Bedroom, and LSUN Outdoor Church. The baseline is trained for 1000 epochs on CIFAR10 with $T = 100$, for 3000 epochs on CIFAR10 with $T = 10$, for 2500 epochs on CelebA, and for 2000 epochs on LSUN Bedroom, and LSUN Outdoor Church. At the beginning of inversion procedure with DEQ-DDIM, we sample $\mathbf{x}_T \sim \mathcal{N}(\mathbf{0}, \mathbf{I})$, and initialize the latents at all (or the subsequence of) timesteps to this value. For inversion with DEQ-sDDIM we also sample $\epsilon_{1:T} \sim \mathcal{N}(\mathbf{0}, \mathbf{I})$. We stop the training as soon as the loss falls below 0.5 for CIFAR10, and below 2 for other datasets.

**Evaluation**   We compute FID scores using the code provided by https://github.com/w86763777/pytorch-gan-metrics. We also use the precomputed statistics for CIFAR10, LSUN Bedrooms and Outdoor Churches provided in this github repository. For CelebA, we compute our own dataset statistics, as the precomputed statistics for images of resolution 64×64 are not included in this repository. While computing the statistics, we preprocess the images of CelebA in exactly the same way as done by Song et al. [68].

# B   Pseudocode

We provide PyTorch-style [60] pseudocode to invert DDIM with DEQ approach in Algorithm box B. Note that we use phantom gradient [28] to compute inexact gradients.

# C   Ablation Studies for Model Inversion with DEQ-DDIM

## C.1   Effect of length of sampling chain

**Effect of length of subsequence $\tau_S$ during training**   We study the effect of the length of the diffusion chain on the convergence rate of optimization for model inversion in Fig. 6. We note that for sequential sampling, loss decreases slightly faster for the smaller diffusion chain ($\tau_S = 10$ and 20) than the longer one ($\tau_S = 100$) for the baseline. However, for DEQs, the length of diffusion chain doesn't seem to have an effect on the rate of convergence as the loss curves for $\tau_S = 100$ and $\tau_S = 10$ and 20 are nearly identical.

**Effect of length of subsequence $\tau_S$ during sampling**   All the images in Fig. 4 are sampled with a subsequence of timesteps $\tau_S \subset T$, *i.e.,* the number of latents in the diffusion chain used for training and the number of timesteps used for sampling an image from the recovered $\hat{x}_T$ were equal. We investigate if sampling with $\tau_S = T = 1000$ results in images with a better perceptual quality for the baseline. We display the recovered images for LSUN Bedrooms in Fig. 7. The length of diffusion chain during training time is $\tau_S = 10$. We note that using more sampling steps does not result in

**Algorithm 3** PyTorch-style pseudocode for inversion with DEQ-DDIM

```python
# x0: a target image for inversion
# all_xt: all the latents in the diffusion chain or subsequence
# func: A function that performs the required operations on the
# fixed-point system for a single timestep
# solver: fixed-point solver like Anderson acceleration
# optimizer: an optimization algorithm like Adam
# tau: the damping factor τ for phantom gradient
# num_epochs: max number of epochs

def forward(func, x):
    with torch.no_grad():
        z = solver(func, x)
    z = tau * func(z) + (1 - tau) * z
    return z

def invert(func, all_xt, x0, optimizer, num_epochs):
    for epoch in range(num_epochs):
        optimizer.zero_grad()
        xt_pred = forward(func, all_xt)
        loss = (xt_pred[-1] - x0).norm(p='fro')
        loss.backward()
        optimizer.step()
    return all_xt[0]
```

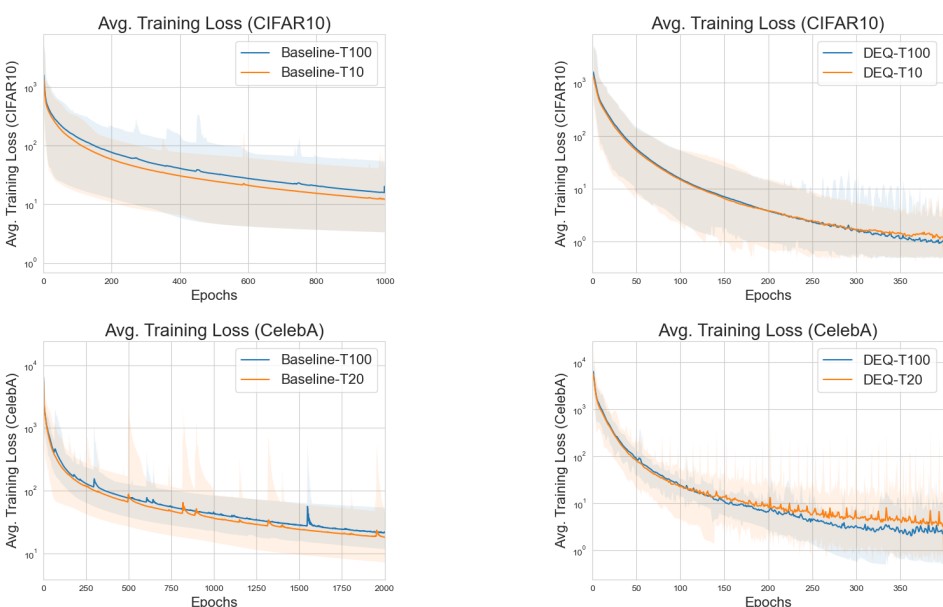

Figure 6: Effect of length of diffusion chain on optimization process for model inversion. with DEQ-DDIM. We display training loss curves for (left) baseline and (right) DEQ for CIFAR10 (top row) and CelebA (bottom row). It is slightly easier to optimize smaller diffusion chain for the Baseline. The error bar indicates the maximum and minimum value of loss encountered during any of the 100 runs for CIFAR10, and 25 runs for CelebA.

inverted images that are closer to the original image. In some cases, samples generated with more sampling steps have some additional artifacts that are not present in the original image.

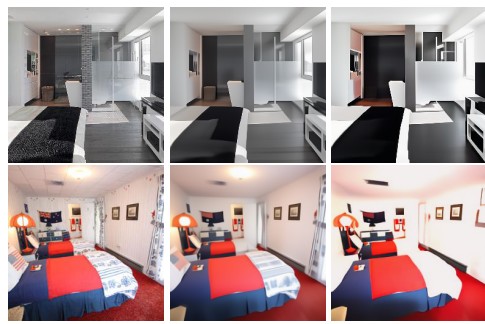 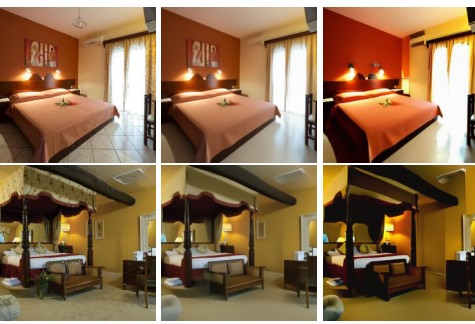

Figure 7: Model inversion on LSUN Bedrooms (Baseline): Using more sampling steps does not generate images that are closer to the ground truth image. Each triplet has the original image (left), images sampled with $T = 10$ (middle) and $T = 1000$ (right). The length of diffusion chain at training time was 10.

**Comparing exact vs inexact gradients for backward pass of DEQ**  The choice of gradient calculation for the backward pass of DEQ affects both the training stability and convergence of DEQs. Here, we compare the performance of the exact gradients and inexact gradients. Computing the inverse of Jacobian in Eq. (15) is difficult because the Jacobian can be prohibitively large. We follow Bai et al. [6] to compute exact gradients using the following linear system

$$\left( J_{g_\theta}^{-1} \big|_{\mathbf{x}_{0:T}^*} \right) \boldsymbol{v}^\top + \left( \frac{\partial \mathcal{L}}{\partial \mathbf{x}_{0:T}^*} \right)^\top = 0 \tag{19}$$

We use Broyden's method [13] to efficiently solve for $\boldsymbol{v}^\top$ in this linear system. We compare it againt inexact gradients *i.e.,* Jacobian free gradient used in Algorithm 2. We observe that training DEQs with exact gradients becomes increasingly unstable as the training proceeds, especially for larger learning rates like 0.005. However, we can converge faster with larger learning rates like 0.01 with inexact gradients.

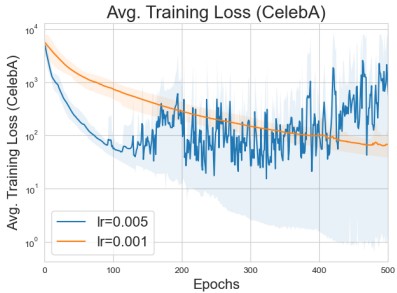 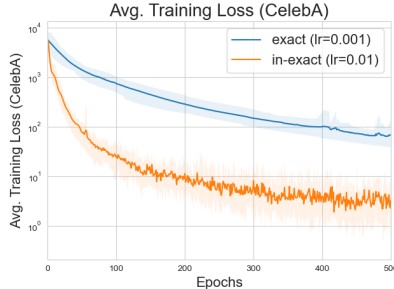

Figure 8: Training DEQs becomes increasingly unstable with exact gradients for larger learning rates like 0.005 (left, blue curve). However, it is possible to train DEQs with exact gradients for smaller learning rate like 0.001 (left, orange curve). However, inexact gradients (right, orange curve) converge faster than exact gradients (right, blue curve) on larger learning rates like 0.01. We report results on 10 runs for CelebA dataset with diffusion chains of length 20.

**Effects of choice of initialization on convergence of DEQs**  The choice of initialization is critical for fast convergence of DEQs. Bai et al. [6] initialize DEQ transformer with zeros. However, in this work, we initialize all the latent states with $\mathbf{x}_T$, as it results in faster convergence shown in Figure 9. While both the initialization schemes converge to high quality images eventually, we observe that initializing with $\mathbf{x}_T$ results in up to $3\times$ faster convergence compared to zero initialization. We observe a significant qualitative difference in the visualization of the intermediate states of the diffusion chain at different solver steps for the two initialization schemes as observed in Figure 10.

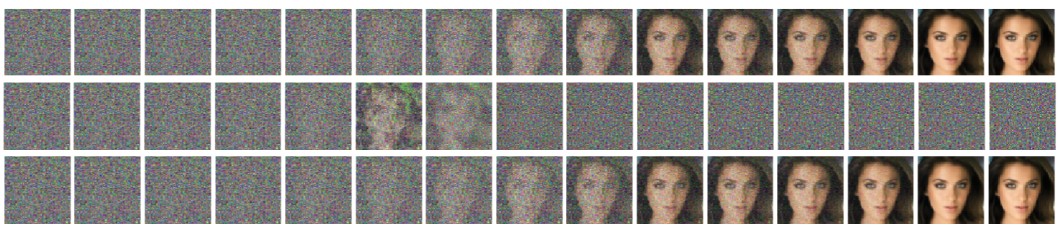

Figure 9: Choice of initialization is critical in DEQs: Initializing DEQs with $\mathbf{x}_T$ results in much faster convergence compared to zero initialization. We report the convergence results on CIFAR10 using 5 runs on diffusion chains of length 50.

Figure 10: Visualization of intermediate latents $\mathbf{x}_t$ for CelebA for different choices of initialization for $T = 50$: (**first row**) Initialization with $\mathbf{x}_T$ after 10 Anderson solver steps (**second row**) Zero initialization after 10 Anderson solver steps (**third row**) Zero initialization after 30 Anderson solver steps. We visualize every $5^{th}$ latent in $\mathbf{x}_{0:T-1}$ at the given solver step (10 or 30). We also visualize $\mathbf{x}_{0:4}$ in the last 5 columns. Initialization with $\mathbf{x}_T$ produces visually appealing images much faster.

## D   Extending DEQ formulation to stochastic DDIM (DEQ-sDDIM)

A more general and highly stochastic generative process is to sample and integrate noises every time step, given by [68]:

$$\mathbf{x}_{t-1} = \sqrt{\alpha_{t-1}}\left(\frac{\mathbf{x}_t - \sqrt{1 - \alpha_t}\boldsymbol{\epsilon}_\theta^{(t)}(\mathbf{x}_t)}{\sqrt{\alpha_t}}\right) + \sqrt{1 - \alpha_{t-1} - \sigma_t^2}\cdot\boldsymbol{\epsilon}_\theta^{(t)}(\mathbf{x}_t) + \sigma_t\boldsymbol{\epsilon}_t \tag{20}$$

where $\epsilon_t \sim \mathcal{N}(\mathbf{0}, \mathbf{I})$. Here, $\sigma_t = 0$ corresponds to a deterministic DDIM while $\sigma_t = \sqrt{\frac{1 - \alpha_{t-1}}{1 - \alpha_t}}\sqrt{1 - \frac{\alpha_t}{\alpha_{t-1}}}$ corresponds to a stochastic DDIM. Empirically, this is parameterized as $\sigma_t(\eta) = \eta\sqrt{\frac{1 - \alpha_{t-1}}{1 - \alpha_t}}\sqrt{1 - \frac{\alpha_t}{\alpha_{t-1}}}$ where $\eta$ is a hyperparameter to control stochasticity. Note that $\eta = 1$ corresponds to a DDPM [67, 33].

Rearranging the terms in Eq. (20), we get

$$\mathbf{x}_{t-1} = \sqrt{\frac{\alpha_{t-1}}{\alpha_t}}\mathbf{x}_t + \left(\sqrt{1 - \alpha_{t-1} - \sigma_t^2} - \sqrt{\frac{\alpha_{t-1}(1 - \alpha_t)}{\alpha_t}}\right)\epsilon_\theta^{(t)}(\mathbf{x}_t) + \sigma_t\boldsymbol{\epsilon}_t \tag{21}$$

Let $c_1^{(t)} = \sqrt{1 - \alpha_{t-1} - \sigma_t^2} - \sqrt{\frac{\alpha_{t-1}(1 - \alpha_t)}{\alpha_t}}$. Then we can write

$$\mathbf{x}_{t-1} = \sqrt{\frac{\alpha_{t-1}}{\alpha_t}}\mathbf{x}_t + c_1^{(t)}\epsilon_\theta^{(t)}(\mathbf{x}_t) + \sigma_t\boldsymbol{\epsilon}_t \tag{22}$$

By induction, we can rewrite the above equation as:

$$\mathbf{x}_{T-k} = \sqrt{\frac{\alpha_{T-k}}{\alpha_T}}\mathbf{x}_T + \sum_{t=T-k}^{T-1} \sqrt{\frac{\alpha_{T-k}}{\alpha_t}} \left( c_1^{(t+1)} \epsilon_\theta^{(t+1)}(\mathbf{x}_{t+1}) + \sigma_{t+1}\boldsymbol{\epsilon}_{t+1} \right), \quad k \in [0,..,T] \quad (23)$$

This again defines a "fully-upper-triangular" inference process, where the update of $\mathbf{x}_t$ depends on the noise prediction network $\epsilon_\theta$ applied to *all* subsequent states $\mathbf{x}_{t+1:T}$.

Following the notation used in the main paper, let $h(\cdot)$ represent the function that performs the operations in the equations (23) for a latent $\mathbf{x}_t$ at timestep $t$, let $\tilde{h}(\cdot)$ represent the function that performs the same set of operations across all the timesteps simultaneously, and let $\boldsymbol{\epsilon}_{1:T}$ represent the noise injected into the diffusion process at every timestep. We can write the above set of equations as a fixed point system:

$$\begin{bmatrix} \mathbf{x}_{T-1} \\ \mathbf{x}_{T-2} \\ \vdots \\ \mathbf{x}_0 \end{bmatrix} = \begin{bmatrix} h(\mathbf{x}_T; \boldsymbol{\epsilon}_T) \\ h(\mathbf{x}_{T-1:T}; \boldsymbol{\epsilon}_{T-1:T}) \\ \vdots \\ h(\mathbf{x}_{1:T}; \boldsymbol{\epsilon}_{1:T}) \end{bmatrix}$$

or,

$$\mathbf{x}_{0:T-1} = \tilde{h}(\mathbf{x}_{0:T-1}; \mathbf{x}_T, \boldsymbol{\epsilon}_{1:T}) \tag{24}$$

The above system of equations represent a DEQ with $\mathbf{x}_T$ and $\boldsymbol{\epsilon}_{1:T}$ as input injection.

A major difference between the both is that DEQ-sDDIM can exploit the noises $\boldsymbol{\epsilon}_{1:T}$ sampled prior to fixed point solving as addition input injections. The insight here is that the noises along the sampling chain are independent of each other, thus allowing us to sample all the noises *simultaneously* and convert a highly stochastic autoregressive sampling process into a deterministic "fully-upper-triangular" DEQ.

Then we can now use black-box solvers like Anderson acceleration [5] to solve for the roots of this system of equations as the previous case. Let $g(\mathbf{x}_{0:T-1}; \mathbf{x}_T, \boldsymbol{\epsilon}_{1:T}) = \tilde{h}(\mathbf{x}_{0:T-1}; \mathbf{x}_T, \boldsymbol{\epsilon}_{1:T}) - \mathbf{x}_{0:T-1}$, then we have

$$\mathbf{x}_{0:T}^* = \text{RootSolver}(g(\mathbf{x}_{0:T-1}); \mathbf{x}_T, \boldsymbol{\epsilon}_{1:T}) \tag{25}$$

where RootSolver$(\cdot)$ is any black-box fixed point solver.

### D.1 Convergence of DEQ-sDDIMs

We verify that our DEQ version for stochastic DDIM converges to a fixed point by plotting values of $\|\tilde{h}(\mathbf{x}_{0:T}) - \mathbf{x}_{0:T}\|_2$ over Anderson solver steps in Figure 11. As one would expect, we need more solver steps to solve for the fixed point given higher values of $\eta$.

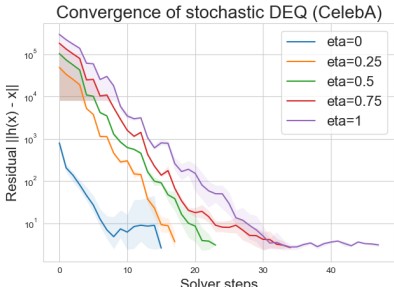
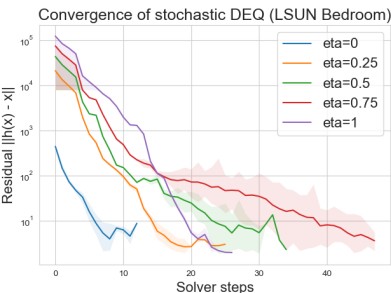

Figure 11: DEQ-sDDIM finds an equilibrium point. We plot the absolute fixed-point convergence $\|\tilde{h}(\mathbf{x}_{0:T}) - \mathbf{x}_{0:T}\|_2$ during a forward pass of DEQ for CelebA (left) and LSUN Bedrooms (right) for different number of steps $T$. The shaded region indicates the maximum and minimum value encountered during any of the 10 runs.

## D.2 Sample quality of images generated with DEQ-sDDIM

We verify that DEQ-sDDIM can generate images that are on par with the original DDIM by computing FID scores on 50,000 sampled images. We report our results in Table 2 and Table 4. We observe that our FID scores are comparable or slightly better to those from sequential DDIM. We also visualize images generated from the same latent $\mathbf{x}_T$ at different levels of stochasticity controlled through $\eta$. We display our generated images in Figure 12 and Figure 13.

| $\eta$ | $T$ | FID Scores | | Time (in seconds) | |
|---|---|---|---|---|---|
| | | DDIM | DEQ-sDDIM | DDIM | DEQ-sDDIM |
| 0.2 | 20 | 13.85 | **13.52** | **0.42** | 1.53 |
| 0.5 | 20 | 15.67 | **15.27** | **0.42** | 2.17 |
| 1 | 20 | 25.85 | **25.31** | **0.42** | 2.35 |
| 0.2 | 50 | 9.33 | **8.66** | **1.05** | 4.17 |
| 0.5 | 50 | 10.75 | **9.73** | **1.05** | 5.18 |
| 1 | 50 | 18.22 | **15.57** | **1.05** | 8.59 |

Table 4: FID scores for single image generation using stochastic DDIM and DEQ-sDDIM on CelebA. Note that DDPM with a larger variance achieves FID score of $183.83^*$ and $71.71^*$ on $T = 20$ and $T = 50$, respectively, where $*$ indicates the numbers from Song et al. [68]

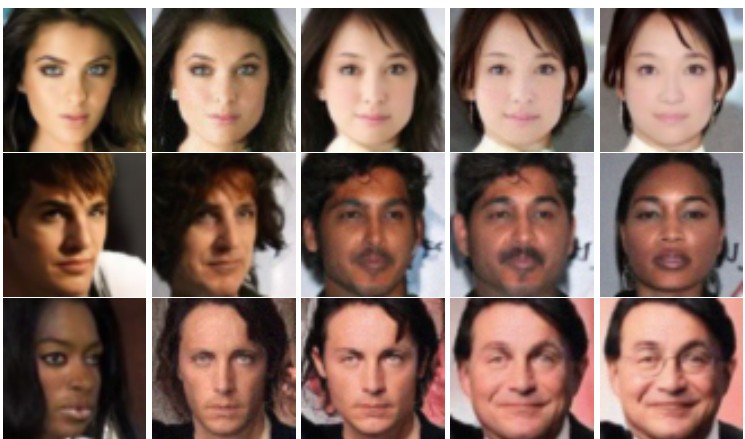

Figure 12: CelebA samples from DEQ-sDDIM for $T = 500$. (**left to right**) We display images from the same $\mathbf{x}_T$ for different levels of stochasticity with $\eta = 0, 0.25, 0.5, 0.75,$ and 1

## D.3 Model inversion with DEQ-sDDIM

We report the minimum values of squared Frobenius norm between the recovered and target images averaged from 25 different runs in Figure 17. We use the same hyperparameters as the ones used for training DEQ models for DDIM in these experiments. DEQ-sDDIM are able to achieve low values of the reconstruction loss even for large values of $\eta$ like 1 as noted in Figure 17. We also plot training loss curves for different values of $\eta$ in Figure 16 on CIFAR10. We note that it indeed takes longer time to invert DEQ-sDDIMs for higher values of $\eta$. However, despite that we obtain impressive model inversion results on CIFAR10 and CelebA. We visualize images generated with the recovered latent states in Figure 14 and Figure 15.

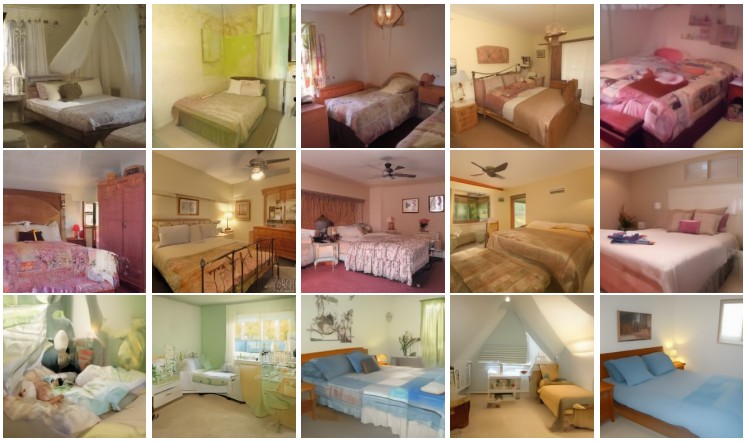

Figure 13: LSUN Bedrooms samples from DEQ-sDDIM for $T = 25$. (**left to right**) We display images from the same $\mathbf{x}_T$ for different levels of stochasticity with $\eta = 0, 0.25, 0.5, 0.75$, and $1$

---

**Algorithm 4** Inverting stochastic DDIM with DEQ (DEQ-sDDIM)

---

**Input:** A target image $\mathbf{x}_0 \sim \mathcal{D}$, $\boldsymbol{\epsilon}_\theta(\mathbf{x}_t, t)$ a trained denoising diffusion model, $N$ the total number of epochs, diffusion function $g$ defined in Eq. (25)

    **Initialize** $\hat{\mathbf{x}}_{0:T} \sim \mathcal{N}(\mathbf{0}, \mathbf{I})$, $\boldsymbol{\epsilon}_{1:T} \sim \mathcal{N}(\mathbf{0}, \mathbf{I})$

    **for** epochs from 1 to $N$ **do**

        $\mathbf{x}^*_{0:T-1} = \text{RootSolver}(g(\mathbf{x}_{0:T-1}); \mathbf{x}_T, \boldsymbol{\epsilon}_{1:T})$.         ▷ Disable gradient computation

        Compute Loss $\mathcal{L}(\mathbf{x}_0, \mathbf{x}^*_0)$.         ▷ Enable gradient computation

        Compute $\partial \mathcal{L} / \partial \mathbf{x}_T$ using 1-step grad.

        Update $\hat{\mathbf{x}}_T$ with gradient descent.

    **end for**

**Output:** $\mathbf{x}^*_T$

---

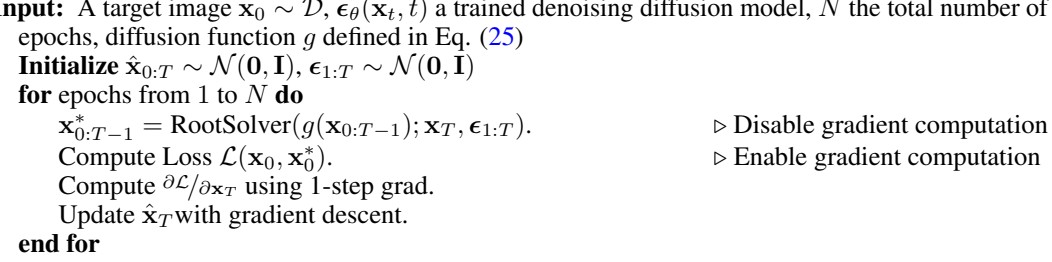

Figure 14: Results of model inversion of DEQ-sDDIM on CIFAR10: For every set of four images from **left to right** we display the original image, and images obtained through inversion for $\eta = 0$, $\eta = 0.5$, and $\eta = 1$.

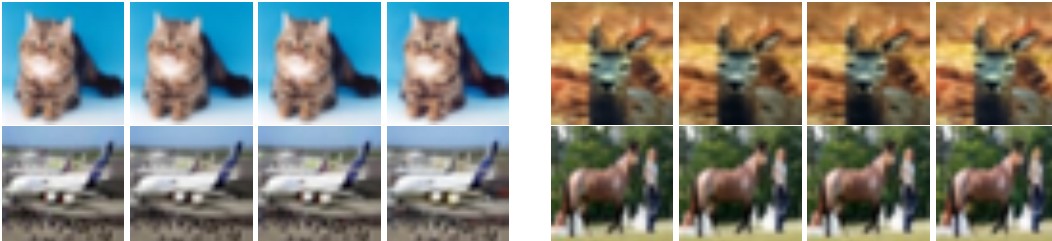

Figure 15: Results of model inversion of DEQ-sDDIM on CelebA: For every set of four images from **left to right** we display the original image, and images obtained through inversion for $\eta = 0$, $\eta = 0.5$, and $\eta = 1$.

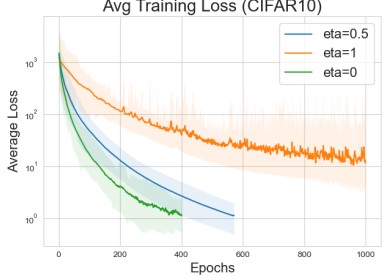

| Dataset | T | $\eta$ | Min loss | Epochs |
|---------|---|--------|----------|--------|
| CIFAR10 | 10 | 0 | $0.76 \pm 0.35$ | 400 |
| CIFAR10 | 10 | 0.5 | $0.49 \pm 0.001$ | 600 |
| CIFAR10 | 10 | 1 | $6.64 \pm 3.45$ | 1000 |
| CelebA | 20 | 0 | $0.41 \pm 0.07$ | 1500 |
| CelebA | 20 | 0.5 | $1.57 \pm 1.63$ | 1500 |
| CelebA | 20 | 1 | $5.03 \pm 1.57$ | 1500 |

Figure 16: Training loss of inversion for DEQ-sDDIM on CIFAR10 averaged over 25 different runs

Figure 17: Minimum loss for inversion with DEQ-sDDIM

# E    Additional Qualitative Results

## E.1    Convergence of DEQ-DDIM for different lengths of diffusion chains

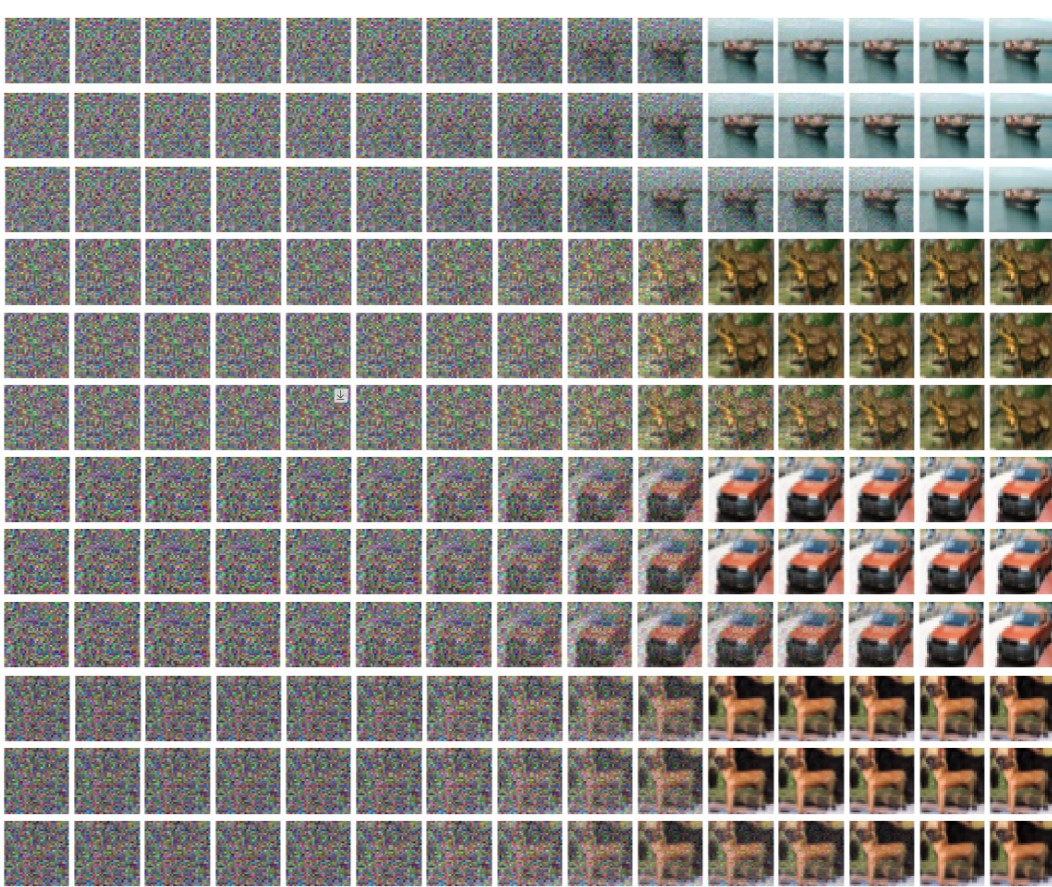

Figure 18: Visualization of intermediate latents $\mathbf{x}_t$ after 15 forward steps with Anderson solver for CIFAR-10. Each set of three consecutive rows displays intermediate latents for different number of diffusion steps: (first row) $T = 1000$, (second row) $T = 500$, and (third row) $T = 50$. For $T = 1000$, we visualize every $100^{th}$, for $T = 500$, we visualize every $50^{th}$ latent, and for $T = 50$, we visualize every $5^{th}$ latent. In addition, we also visualize $\mathbf{x}_{0:4}$ in the last 5 columns.

## E.2 Model inversion with DEQ-DDIM

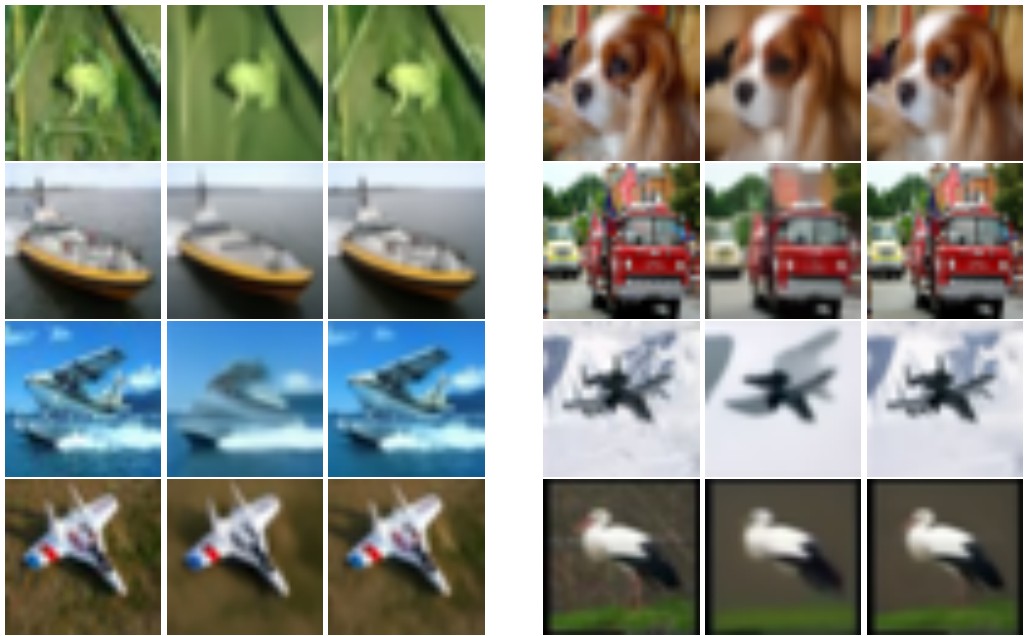

Figure 19: Model inversion on CIFAR10. Each triplet has the original image (left), DDIM's inversion (middle), and DEQ's inversion (right). The number of sampling steps for all the images is $T = 100$.

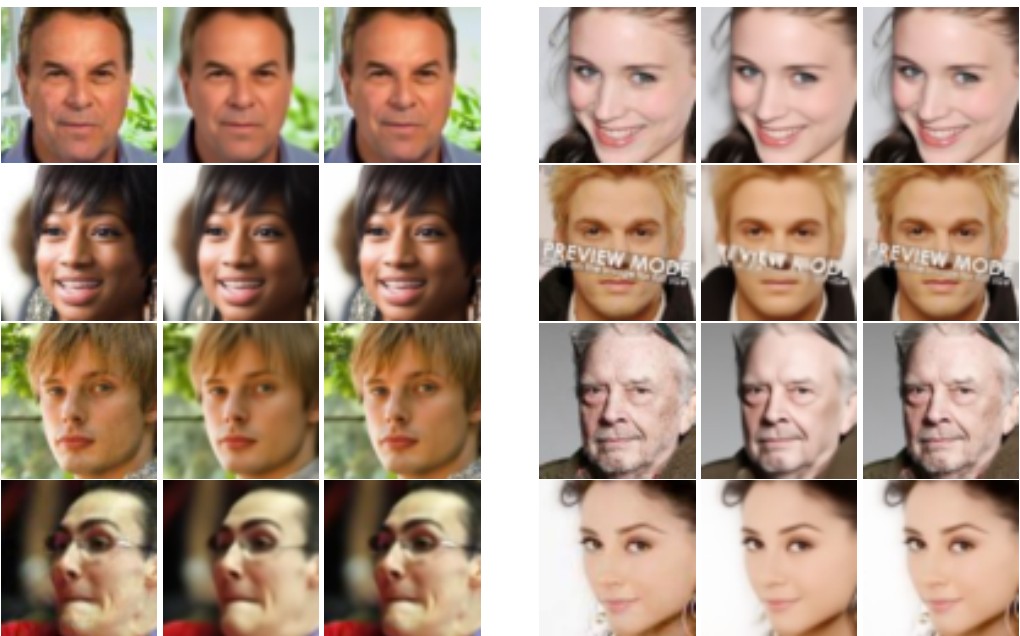

Figure 20: Model inversion on CelebA. Finer details like hair, background, and texture of skin are better captured by DEQ. Each triplet has the original image (left), DDIM's inversion (middle), and DEQ's inversion (right). The number of sampling steps for all the generated images is $T = 10$.

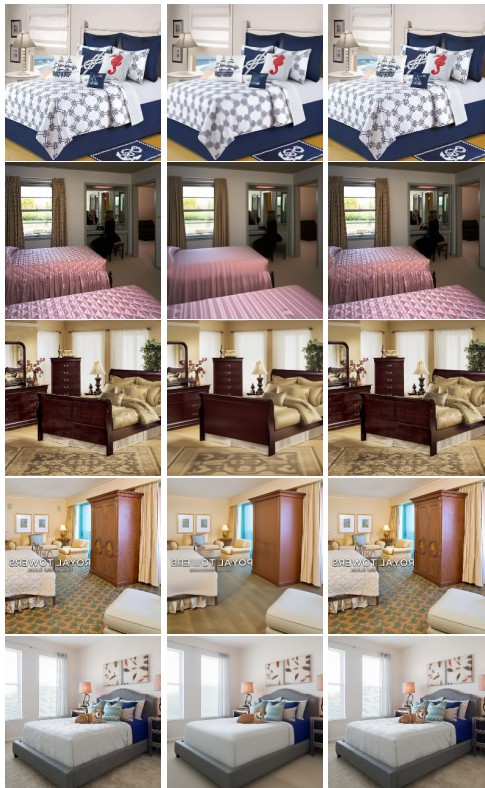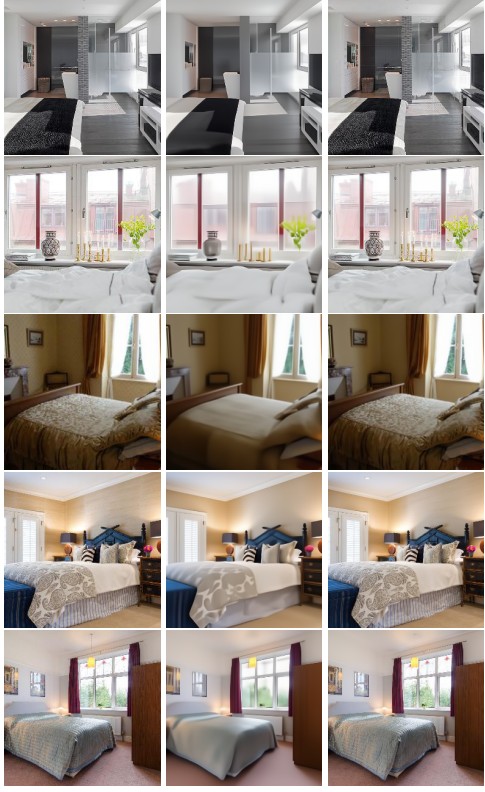

Figure 21: Model inversion on LSUN Bedrooms. Each triplet has the original image (left), DDIM's inversion (middle), and DEQ-DDIM's inversion (right). The number of sampling steps for all the generated images is $T = 10$.

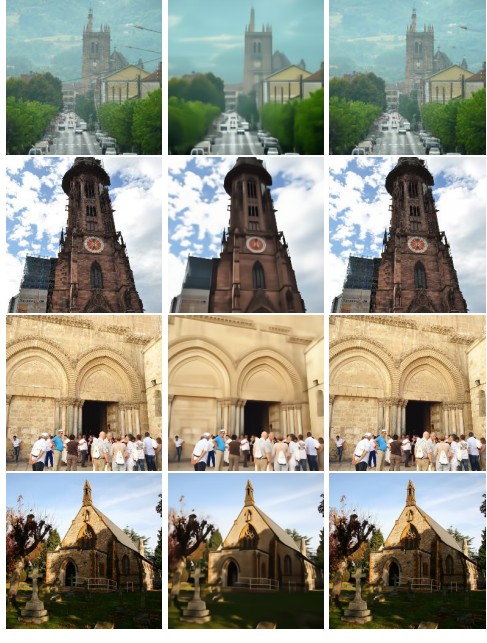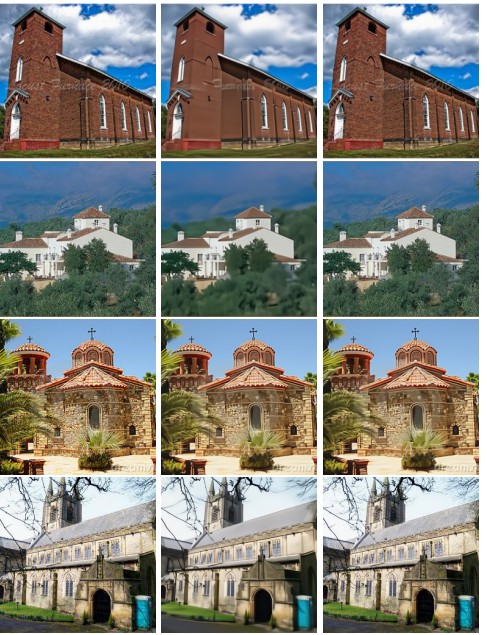

Figure 22: Model inversion on LSUN Outdoor Churches. Each triplet has the original image (left), DDIM's inversion (middle), and DEQ-DDIM's inversion (right). The number of sampling steps for all the generated images is $T = 10$.

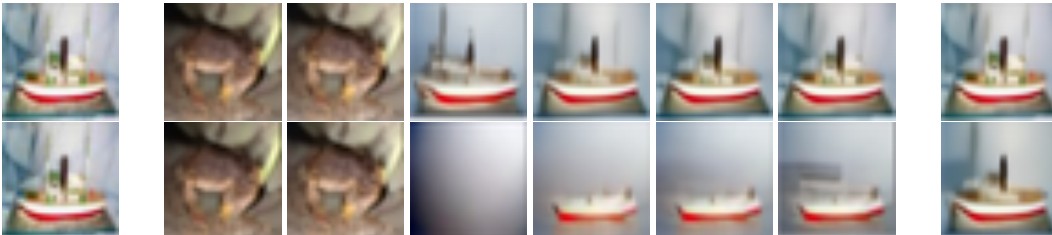

Figure 23: Visualization of model inversion on CIFAR10. The first column is the original image and the last column is the final generated image after 1000 steps for the baseline, and 500 steps for DEQ-DDIM. The 6 columns in between contain images sampled from $\mathbf{x}_T$ after $n$ training updates where $n$ is 0 (initialization), 1, 50, 100, 150, and 200 for DEQ-DDIM (first row) and baseline (second row).

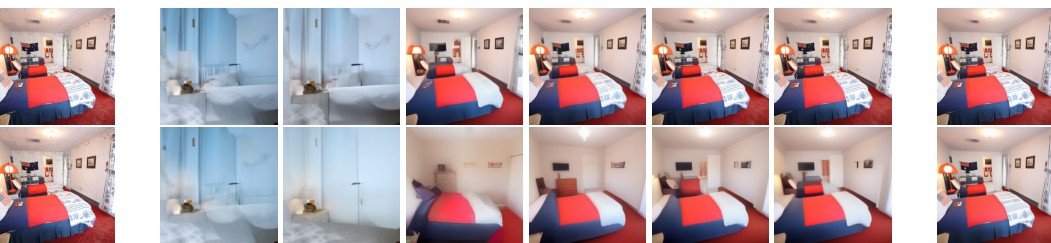

Figure 24: Visualization of model inversion on LSUN Bedrooms. The first column is the original image and the last column is the final generated image after 2000 steps for the baseline, and 500 steps for DEQ-DDIM. The 6 columns in between contain images sampled from $\mathbf{x}_T$ after $n$ training updates where $n$ is 0 (initialization), 1, 50, 100, 150, and 200 for DEQ-DDIM (first row) and baseline (second row).

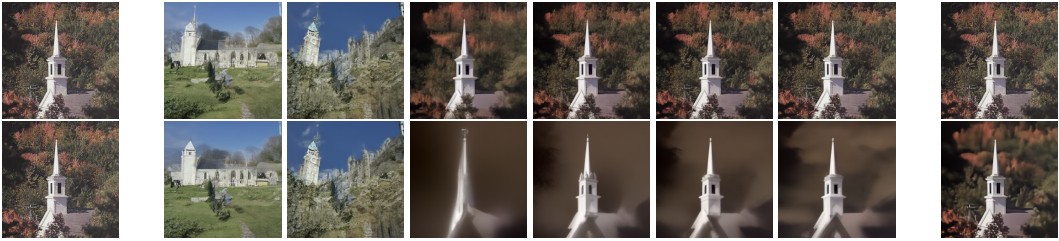

Figure 25: Visualization of model inversion on LSUN Churches. The first column is the original image and the last column is the final generated image after 2000 steps for the baseline, and 500 steps for DEQ-DDIM. The 6 columns in between contain images sampled from $\mathbf{x}_T$ after $n$ training updates where $n$ is 0 (initialization), 1, 50, 100, 150, and 200 for DEQ-DDIM (first row) and baseline (second row).