# OpenReview forum: "Deep Equilibrium Approaches to Diffusion Models"
_NeurIPS.cc/2022/Conference — NeurIPS 2022 Accept_

### Official Review · Reviewer_7km1 · 2022-07-08

**Rating:** 7
**Confidence:** 4
**Soundness:** 4 excellent
**Presentation:** 2 fair
**Contribution:** 3 good

**Summary:**

The paper shows a deep equilibrium interpretation of denoising diffusion implicit models (DDIM). This approach provides the traditionally sequential diffusion models with parallelization capabilities, resulting in faster single image generation and model inversion with less memory consumption.

**Questions:**

- Is algorithm 1 the same as the inversion method suggested in DDIM (section 5.4 of DDIM's arXiv version)?

- What is the reasoning behind the choices of T for different datasets?

---

Post-rebuttal:

The authors have sufficiently answered my questions and concerns, and have shown willingness to provide more experiments in the final version. I have thus raised my score to 7.

**Limitations:**

The paper holds no potential negative societal impact in my opinion, and the authors have aptly ignored this as it is not applicable.

As for the limitations, the authors touch upon their method's limitations in the paper text, addressing the larger memory requirements of each step. However, DDIM's results should be made clearer. It should also be mentioned that sequential models can be differentiated through with constant memory complexity as well (yes, current autograd methods cannot automatically achieve this, but it still is possible).

**Strengths And Weaknesses:**

Strengths:
- Well-founded and strong mathematical viewpoints on diffusion models. This paper provides a novel and significant view on diffusion models as upper-triangular iterative systems, and shows an interesting method to differentiate through them.
- Impressive model inversion capabilities, especially as demonstrated qualitatively in Figure 4.


Weaknesses:
- Line 53: "modern autograd packages would require storing the entire computational graph for all T states by default". While true, it is still possible to differentiate through the entire sequential diffusion process without requiring this. For example, this was done in [R1]. This fact is not made sufficiently clear to the reader. The same point is made on line 159, without stating "by default".
- The approach loses its appeal when considering batched image generation.
- In Table 1, the reporting of DDIM results is inadequate. DDIM achieves an FID of 4.07 using a particular setting, whereas it can also achieve an FID of 3.17 using a different parameterization, under the same timeframe. The fact that it edges out this paper's result is okay, as this paper provides a significant speedup. However, this result should be adequately reported in the table.
- Similarly, DDIM can be sped up significantly and achieves remarkable results. The authors suggest that their method can be used in tandem with other speedup methods, which sounds theoretically plausible, but such results are not reported in the paper.
- Results seem to only mildly improve upon previous results seen in DDIM and others. An additional experiment showing the use of both the novel acceleration method and previous speedup methods simultaneously would better demonstrate the capabilities of this method.

Minor comments:
- Typo on line 41: "join" instead of "joint"
- Typo on line 209: "effect" instead of "affect"
- On lines 266-268, some citations are not described correctly. Citations [45, 17, 57, 32] do not manipulate latent codes to edit high-level attributes of images. They utilize the prevalent understanding of diffusion models for the respective tasks they solve, similar to ideas first presented in [57, R2, R3]. The text should be reworded and show the existing and suggested citations as methods that edit images or solve inverse problems without requiring full model inversion.

[R1] Nie, W., Guo, B., Huang, Y., Xiao, C., Vahdat, A., & Anandkumar, A. (2022). Diffusion Models for Adversarial Purification. arXiv preprint arXiv:2205.07460.

[R2] Kadkhodaie, Z., & Simoncelli, E. P. (2020). Solving linear inverse problems using the prior implicit in a denoiser. arXiv preprint arXiv:2007.13640.

[R3] Kawar, B., Vaksman, G., & Elad, M. (2021). SNIPS: Solving noisy inverse problems stochastically. Advances in Neural Information Processing Systems, 34, 21757-21769.

---

> ### Author Response · Authors · 2022-08-02
> **Response to Reviewer 7km1 (2/2)**
>
> > _Similarly, DDIM can be sped up significantly and achieves remarkable results. The authors suggest that their method can be used in tandem with other speedup methods, which sounds theoretically plausible, but such results are not reported in the paper._
> > _Results seem to only mildly improve upon previous results seen in DDIM and others. An additional experiment showing the use of both the novel acceleration method and previous speedup methods simultaneously would better demonstrate the capabilities of this method._
>
> We are sorry for not expanding this point in the main paper. Progressive distillation by Salimans and Ho, 2022, [R1] is an example of a work that is complementary to our approach. A DDIM/DDPM model that is distilled through progressive distillation can achieve further speed up with a DEQ. Due to strict timelines, we were unable to provide additional results on this in the current draft. However, we are happy to explore this direction and include additional results in the final draft.
>
> [R1] Salimans, Tim, and Jonathan Ho. "Progressive distillation for fast sampling of diffusion models." International conference on learning representations (2022).
>
> > _On lines 266-268, some citations are not described correctly. Citations [45, 17, 57, 32] do not manipulate latent codes to edit high-level attributes of images. They utilize the prevalent understanding of diffusion models for the respective tasks they solve, similar to ideas first presented in [57, R2, R3]. The text should be reworded and show the existing and suggested citations as methods that edit images or solve inverse problems without requiring full model inversion._
>
> Thank you for pointing us to additional works that are extremely relevant and very informative. We have reworded the text and have cited the suggested works.
>
> > _Is algorithm 1 the same as the inversion method suggested in DDIM (section 5.4 of DDIM's arXiv version)?_
>
> Algorithm 1 differentiates through the diffusion chain to minimize reconstruction error over multiple steps. In contrast, Sec 5.4 in the DDIM’s paper performs reconstructing using only a _single_ forward and backward pass in the model, and doesn’t perform any optimization to further reduce it, which is precisely what our method focuses on.
>
> > _What is the reasoning behind the choices of T for different datasets?_
>
> * **For single image generation**: In an ideal case, we should observe significant gains in speedup over sequential sampling for DEQs provided we can fit the entire batch of the diffusion sampling chain in the memory and use a good fixed-point solver. For instance, as noted in Figure 17,, for T=1000 on CIFAR10, sequential sampling would require T=1000 steps but with Anderson solver we can get good quality samples in as few as 15 steps. However, for images with larger resolutions, we found it difficult to fit large batches on GPUs. Therefore, we only report results on shorter diffusion chains for larger images.
>
> * **For model inversion**: We used shorter diffusion chains for model inversion because empirically we did not notice significant gains in convergence rate and reconstruction loss with longer diffusion chains (Figure 5 in supplementary material). Further, longer diffusion chains are slower to optimize. Therefore we only included results on shorter diffusion chains in the main paper.

---

> ### Author Response · Authors · 2022-08-02
> **Response to Reviewer 7km1 (1/2)**
>
> We are grateful for a thorough review and extremely constructive feedback! Thank you for stating that this work offers ‘well-founded and strong mathematical viewpoints on diffusion models’.
>
> > _Line 53: "modern autograd packages would require storing the entire computational graph for all T states by default". While true, it is still possible to differentiate through the entire sequential diffusion process without requiring this. For example, this was done in [R1]. This fact is not made sufficiently clear to the reader. The same point is made on line 159, without stating "by default"._
>
> We apologize for not highlighting other methods that can differentiate through the sequential processes efficiently. We have updated our draft per your suggestion. Specifically, we have cited Nie et. al. 2022 as a recent work that also differentiates through the diffusion process efficiently through SDE solvers. We have also deleted Line 159.
>
> > _In Table 1, the reporting of DDIM results is inadequate. DDIM achieves an FID of 4.07 using a particular setting, whereas it can also achieve an FID of 3.17 using a different parameterization, under the same timeframe. The fact that it edges out this paper's result is okay, as this paper provides a significant speedup. However, this result should be adequately reported in the table._
>
> Thank you for pointing this out. We have **updated Table 1** to include results for DDPM. We also propose a simple way to extend our current framework to include stochastic DDIM. The key idea is to sample noise for all the time steps and treat this noise as an input injection to DEQ, in addition to $x_T$. Thus, our current model can be **seamlessly extended to integrate injected noise**. We have provided a derivation of this formulation in **Section D** of supplementary material. Our preliminary results on CelebA indicate that DEQ formulation of stochastic DDIM (DEQ-sDDIM) can achieve FID scores that are on par with or better than those of sequential DDIM (Table 3). Further, different levels of stochasticity can generate diverse images from the same initial latent $x_T$ (Figure 11 and 12). Finally, we obtain impressive results on model inversion with DEQ-sDDIM on CelebA and CIFAR10 (Figure 13, 14). Due to tight timelines during the rebuttal period we were unable to perform more extensive experiments but we promise to include more extensive empirical  results in the final version. Please let us know if you have any additional questions about this formulation.

---

### Official Review · Reviewer_cuLf · 2022-07-11

**Rating:** 5
**Confidence:** 4
**Soundness:** 2 fair
**Presentation:** 3 good
**Contribution:** 3 good

**Summary:**

This paper develops a variant (deep equilibrium model (DEQ)) of an existing method (DDIM) with the goal of efficient sampling and model inversion while still maintaining the performance.  In particular, it treats the sampling step sequence of DDIM（denoising diffusion implicit model, where the sampling step is deterministic and no longer a Markov chain.) as a fixed-point set, so that a fixed-point solver (e.g., Anderson acceleration, etc.) is able to jointly minimize the entire sampling chain. The authors show the practical relevance of their proposal in single-shot image sampling and model inversion on several benchmark datasets.

**Questions:**

1), From the high-level point of view, what is the practical meaning of single-shot image sampling? From my perspective, I think image restoration and reconstruction (e.g., Inpainting, super-resolution, medical image inverse problems, etc.) might be a potential direction. However, as mentioned by authors, the proposal may work efficiently only on shorter sampling lengths, and on images with smaller resolutions (32x32, 64x64). As a result, the rationale for single sampling need to be discussed from very beginning.

2), In the same line above, there is supposed to have an original DDPM model for references in all the experiments, as presented in [Song et al., 2021], DDPM models works better than DDIM when sampling steps turn to be larger. The addition of these comparisons will clearly show the regime that suits to this proposal.

3), In line 146-147, the statement, “Solving for all the equilibria simultaneously leads to a better estimation of the intermediate latent states x_t in a fewer number of steps (i.e., $≤ t$ steps for $x_t$).”, is confusing and need to be justified with some ablation studies. The jointly sampling still need to process all the T steps, right?

4), In this work, it is claimed that the steps can be executed as mini-batched in parallel. Is the parallel computing already used at DDIM sampling, such as the outcomes in Table 1 and other experiments? If so, please consider adding more implementation details in the revision. Some additional explanation through Eq. (7) would be helpful. In addition, will the parallel computing work in the multi-batch cases?

5), What is the initialization $x_ {0: T-1}$ of the fixed-point iterations?  In the original DEQ implementations, the initialization is usually set to zero. Will different initializations result in different convergence and sampling performance?

6), I’m curious about how this proposal can be applied to a more general diffusion models that the sampling step is not deterministic. Seems that the current methodological statements in Sec. 3, from Eq. 9 to Eq. 10, are not clear for stochastic sampling. It would be very helpful for authors to discuss this point.

7), When comparing with DDIM, only LSUN datasets with resolution 256x256 are considered. As exploring advances in large scale diffusion models for image generation is impactful, I wonder how the proposed jointly sampling compares to DDIM on ImageNet of resolution 512x512? The pretrained models are publicly available in https://github.com/openai/guided-diffusion.

8), Since the proposed DEQ method is mainly based on diffusion sampling [30] and DDIM sampling [54], how the architecture improvements apply to the SDE version of score-based models and the corresponding conditional SDEs?

9), The current gradients calculation of model inversion is inexact due to the usage of Jacobian free. I wonder the current proposal can fed into the exact gradient calculation for the backward pass (e.g., Neumann backpropagation, etc.).


**Limitations:**

The limitations of the proposal are missing or at least not well stated in the current manuscript.

**Strengths And Weaknesses:**

This paper builds on other works in the recent literature and proposes something useful and novel. It is overall well written and reports (somewhat) competitive results with both quantitative and qualitative validation for image sampling and model inversion.  While I think the methodological aspect is novel enough for publication, the rationale for single image sampling is unclear and need to be further justified. Please find my technical comments bellow.

---

> ### Author Response · Authors · 2022-08-02
> **Response to Reviewer cuLf (3/3)**
>
> > _Since the proposed DEQ method is mainly based on diffusion sampling [30] and DDIM sampling [54], how the architecture improvements apply to the SDE version of score-based models and the corresponding conditional SDEs?_
>
> Extending DEQs to score based sampling (SDEs) is certainly an interesting direction. Designing DEQs needs careful thinking and implementation. Currently we are unsure if our approach can be seamlessly extended to SDE version of score-based models and we will leave this as future work.
>
> > _The current gradients calculation of model inversion is inexact due to the usage of Jacobian free. I wonder if the current proposal can be fed into the exact gradient calculation for the backward pass (e.g., Neumann backpropagation, etc.)._
>
> We use inexact gradients because they are very light weight to compute and stable. We experimented with exact gradients by computing the inverse Jacobian through a linear system as proposed in Bai et. al. 2019 [1] but this method leads to stability issues due to ill-conditioning of Jacobian during the optimization process [2]. We include a plot of the training curve (Figure 7) to compare the performance of exact vs inexact gradients in **Section C** of the supplementary material. Empirically, we observe that we can achieve **faster convergence with inexact gradients at larger learning rates**, while the **exact gradient computed through implicit function theorem can suffer from training stability issues** in this learning rate region.
>
> [1] Bai, Shaojie, J. Zico Kolter, and Vladlen Koltun. "Deep equilibrium models." Advances in Neural Information Processing Systems 32 (2019).
>
> [2] Bai, Shaojie, Vladlen Koltun, and J. Zico Kolter. "Stabilizing equilibrium models by jacobian regularization." arXiv preprint arXiv:2106.14342 (2021).

---

> > ### Comment · Reviewer_cuLf · 2022-08-08
> > **Thank you for your response.**
> >
> > The reviewer would like to first thank the authors for their great efforts in both revising the manuscript and providing a conclusive response. The authors put great effort into addressing my concerns. However, the reviewer still doubts the practical usage of this proposal for small image generatiation, which makes the contributions somewhat shrink to model inversion only. As a result, I think this work is on the borderline of this year's NeurIPS and I would like to maintain my borderline acceptance.

---

> ### Author Response · Authors · 2022-08-02
> **Response to Reviewer cuLf (2/3)**
>
> > _In this work, it is claimed that the steps can be executed as mini-batched in parallel. Is the parallel computing already used at DDIM sampling, such as the outcomes in Table 1 and other experiments? If so, please consider adding more implementation details in the revision. Some additional explanation through Eq. (7) would be helpful. In addition, will the parallel computing work in the multi-batch cases?_
>
> We leverage PyTorch’s inbuilt DataParallel module to achieve parallelism. This function takes care of splitting large batches of inputs across multiple GPUs. Multi-batch cases have been accounted for through careful coding. For the quantitative results reported in the main paper, we use anywhere between 1-4 GPUs NVIDIA RTX A6000 GPUs. We have added these details in the supplementary material. We also plan to release our code along with the paper so that the exact implementation details are available to the community.
>
> >  _What is the initialization $x\_{0:T−1}$ of the fixed-point iterations? In the original DEQ implementations, the initialization is usually set to zero. Will different initializations result in different convergence and sampling performance?_
>
> Indeed, correct initialization is critical in DEQs for better convergence and stability, and different initializations will lead to different rates of convergence. We initialize $x_{0:T-1}$ to $x_T$ as this choice is intuitive and works the best in practice. **Alternate initialization like zero initialization works too but needs more solver steps to achieve convergence.** We provide few quantitative and qualitative results to compare zero initialization vs $x_T$ initialization. We find that zero initialization roughly takes three times as many solver steps  to converge compared to $x_T$ initialization. We have added these plots and some qualitative results in **Section C** of the supplementary material, specifically Figures 8 and 9.
>
> >  _I’m curious about how this proposal can be applied to more general diffusion models that the sampling step is not deterministic. Seems that the current methodological statements in Sec. 3, from Eq. 9 to Eq. 10, are not clear for stochastic sampling. It would be very helpful for authors to discuss this point._
>
> The current DEQ framework can be **extended to the cases where the generative process is non-stochastic**. The key idea is to sample noise for all the time steps and treat this noise as an input injection to DEQ, in addition to $x_T$. Thus, our current model can be **seamlessly extended to integrate injected noise**. Our preliminary results on CelebA indicate that DEQ formulation of stochastic DDIM (DEQ-sDDIM) can achieve FID scores that are on par with or better than those of sequential DDIM (Table 3). Further, different levels of stochasticity can generate diverse images from the same initial latent $x_T$ (Figure 11 and 12). Finally, we obtain impressive results on  model inversion on CelebA and CIFAR10 (Figure 13, 14). We have provided details of this more general DEQ formulation in **Section D**  of the supplementary material. We implemented this extension during the rebuttal week. Due to tight timelines, we provide only preliminary empirical results. Our results are encouraging and we will include more extensive analysis in the final version. Thank you for your valuable comments!
>
> > _When comparing with DDIM, only LSUN datasets with resolution 256x256 are considered. As exploring advances in large scale diffusion models for image generation is impactful, I wonder how the proposed jointly sampling compares to DDIM on ImageNet of resolution 512x512? The pretrained models are publicly available in https://github.com/openai/guided-diffusion._
>
> Our experiments were limited by the capacity of GPUs available to us. Training diffusion models from scratch to achieve good FID is tough due to constraints of compute resources. We therefore used  pre-trained checkpoints as much as possible. Using pretrained checkpoints from  https://github.com/openai/guided-diffusion is difficult as our codebase is very different from their codebase. Due to limited time available to us, merging our codebases is also tricky. We will try to include ImageNet results, especially for model inversion, in the final version.

---

> ### Author Response · Authors · 2022-08-02
> **Response to Reviewer cuLf (1/3)**
>
> Thank you for carefully reviewing this work and asking us very insightful questions! We have tried our best to answer your questions below.
>
> > _From the high-level point of view, what is the practical meaning of single-shot image sampling? From my perspective, I think image restoration and reconstruction (e.g., Inpainting, super-resolution, medical image inverse problems, etc.) might be a potential direction. However, as mentioned by authors, the proposal may work efficiently only on shorter sampling lengths, and on images with smaller resolutions (32x32, 64x64). As a result, the rationale for single sampling needs to be discussed from the very beginning._
>
> We apologize for the confusing terminology. We have updated “single shot image generation” to “single image generation” in our draft; the only point we are making here is that we are generating a _single_ image rather than a batch of images. Indeed, you have rightly pointed out that single image generation has applications in inpainting, image restoration, and solving inverse problems (this is probably the most common use case for interacting with such models). We also use single image generation in our experiments for model inversion. In the real world, manipulation on a single image is certainly a common use-case. As per your suggestions, we have also updated the introduction to include a brief motivation for single image generation. (Line 49-50)
>
> We would also like to highlight that single image generation is just one aspect of this model. The primary strength of this proposed model is *the ability to differentiate through the diffusion chain efficiently*, which gives us a powerful technique to invert these models. For example, we can efficiently invert images of sizes 256X256 and achieve low reconstruction error.
>
> Finally, we would like to address your concern that the proposed method might work only on shorter diffusion chains and smaller image resolutions.
> - For model inversion, we use shorter diffusion chains because we did not observe significant gains in convergence rate and reconstruction loss with longer diffusion chains (Figure 5 in supplementary material). Further, longer diffusion chains are slower to optimize. Therefore we only included results on shorter diffusion chains in the main paper.
> - For single image generation, we hope to clarify that the performance gains from DEQs would likely be more visible on longer diffusion chains, provided we can fit the entire diffusion chain on GPUs, and use a sufficiently powerful fixed-point solver. As noted in Figure 2, for T=500, sequential sampling would require T=500 steps, but with the Anderson solver, we can get good quality samples in as few as 15 steps.
>
> > _In the same line above, there is supposed to have an original DDPM model for references in all the experiments, as presented in [Song et al., 2021], DDPM models works better than DDIM when sampling steps turn to be larger. The addition of these comparisons will clearly show the regime that suits to this proposal._
>
> We have updated Table 1 to include the original DDPM results. We also have some preliminary results for single image generation and  model inversion with the DEQ version of stochastic DDIM (DEQ-sDDIM) that we have included in **Section D** of  the supplementary material.
>
> > _In line 146-147, the statement, “Solving for all the equilibria simultaneously leads to a better estimation of the intermediate latent states $x_t$ in a fewer number of steps (i.e., $\leq t$ steps for $x_t$).”, is confusing and need to be justified with some ablation studies. The joint sampling still needs to process all the $T$ steps, right?_
>
> In context of DEQs, the steps in this statement refer to the **solver steps** and not the actual time steps $t$. Due to the upper-triangular dependency of the inference process, we have better estimates of the latent states which leads to faster convergence of the diffusion chain. This phenomenon can be qualitatively seen in Figure 2 of the main paper and in Figure 17 of the supplementary material. As shown in Figure 17, **diffusion chains as long as 1000 can converge in as few as 15 solver steps**. In contrast, sequential sampling would need 1000 steps for a comparable perceptual quality. It is indeed true that joint sampling needs to maintain $T$ latents $x_{0:T-1}$ as an equilibrium *state*; sampling requires fewer than $T$ steps. This is possible because we solve for the joint equilibria through solvers like Anderson acceleration which leads to faster convergence in a fewer number of steps.

---

### Official Review · Reviewer_MgS2 · 2022-07-12

**Rating:** 6
**Confidence:** 2
**Soundness:** 2 fair
**Presentation:** 3 good
**Contribution:** 2 fair

**Summary:**

This paper proposes an extension to "denoising diffusion implicit model" (DDIM), a variant of denoising diffusion probabilistic model (DDPM). Here I refer to it as DEQ-DDIM. Specifically, it regards the denoising iterations as fixed-point iterations and solves all the denoised images efficiently using Anderson acceleration. This is an interesting exploration; however I have concerns of the reduced expressiveness brought about by DDIM and DEQ-DDIM.


**Questions:**

1. I understand that in the DDIM and this paper, the authors use $\alpha$ to denote $\bar{\alpha}$ in the DDPM paper. However, the authors are suggested to explicitly point it out to avoid possible confusions of readers.
2. Typos and grammar:

a) line 41: join equilibria -> joint equilibria

b) line 137: compared to the traditional -> in contrast to the traditional


**Strengths And Weaknesses:**

Strengths:
1. It's a novel and interesting perspective to view the denoising iterations as fixed-point iterations (seeking for converged solutions through iterations).

Weaknesses:
1. DEQ-DDIM inherits the limitations of DDIM: they remove the injected random noises during denoising, therefore the results are deterministic (given the noise image x_T). This limitation is fundamental: it makes DDIM and variants less being probabilistic models. A direct consequence is the generated images of DDIM and variants has reduced diversity compared with DDPM (refer to the recall of DDIM in [a]). It's debatable whether such a sacrifice for speedup is worthy.
~~2. Another possible inherent flaw (but I might be wrong) of DEQ-DDIM is, as the authors correctly pointed out, "the diffusion process is not time-invariant (i.e., not 'weight-tied' in the DEQ sense)". More specifically, the denoising function takes the (time embedding of) time t as extra input. In my view, the time t controls the variance of the noises to be estimated, therefore it's indispensable. But if taking time t as extra input, then can we still view DDIM as fixed-point iterations? If not, then speedup using Anderson acceleration is not applicable.~~

[a] Tackling the Generative Learning Trilemma with Denoising Diffusion GANs. ICLR 2022.

---

> ### Author Response · Authors · 2022-08-02
> **Response to Reviewer MgS2**
>
> Thank you for your valuable comments and suggestions! We would like to first address your major concerns regarding this work:
>
> > _DEQ-DDIM inherits the limitations of DDIM: they remove the injected random noises during denoising, therefore the results are deterministic (given the noise image $x_T$). This limitation is fundamental: it makes DDIM and variants less probabilistic models. A direct consequence is the generated images of DDIM and variants has reduced diversity compared with DDPM (refer to the recall of DDIM in [a]). It's debatable whether such a sacrifice for speedup is worthy._
>
> This is a good question but there are additional aspects we should consider while evaluating a model. Speed and quality are trade-offs in some circumstances. In our case, we show that it is possible to generate images with *comparable FID scores* to DDPM while offering additional benefits of *differentiating and tuning the entire sampling chain* via DEQ. For example, we can very efficiently invert the sampled image in the latent space. This is what DDIM and DDPM cannot easily accomplish.  Therefore, we think that this is *not* necessarily a "sacrifice for speedup"; the combination between DEQ and diffusion models naturally provides us several choices, including speedup, latent space manipulation, etc.
>
> **Extending DEQ to stochastic DDIM**: We have also taken your concern about the lack of injected noises into account and have worked on a preliminary version of stochastic DDIM (DEQ-sDDIM)  that accounts for the injected noise.  We have added this new formulation in **Section D** of the supplementary material. The key idea is to sample noise for all the time steps along the sampling chain and treat this noise as an input injection to DEQ, in addition to x_T. Thus, our current model can be **seamlessly extended to integrate injected noise**. Our preliminary results on CelebA  indicate that the DEQ for stochastic DDIM can achieve FID scores that are on par with or better than those of sequential DDIM (Table 3). Further, different levels of stochasticity can generate diverse images from the same initial latent $x_T$ (Figure 11 and 12). Finally, we obtain impressive results on model inversion with DEQ-sDDIM on CelebA and CIFAR10 (Figure 13, 14).  Due to tight timelines during the rebuttal period we were unable to perform more extensive experiments but we promise to include more extensive analysis in the final version. Please let us know if you have any additional questions about this formulation.
>
>
> > _Another possible inherent flaw (but I might be wrong) of DEQ-DDIM is, as the authors correctly pointed out, "the diffusion process is not time-invariant (i.e., not 'weight-tied' in the DEQ sense)". More specifically, the denoising function takes the (time embedding of) time t as extra input. In my view, the time t controls the variance of the noises to be estimated, therefore it's indispensable. But if taking time t as extra input, then can we still view DDIM as fixed-point iterations? If not, then speedup using Anderson acceleration is not applicable._
>
> We hope to clarify that the DEQ-DDIM actually takes the entire **sequence** of time-dependent DDIM as a fixed point system, which means the temporal dimension $T$ is parallelly embedded into the DEQ-DDIM system. DEQ takes in the entire sequence $x_{T-1},...,x_0$ as the state, and $x_T$ as input injection (resulting in a system of $T$ fixed point equations) and solves for the joint equilibria of the entire system of equations. Indeed, the sequential sampling is dependent on time, and therefore cannot be viewed as a fixed-point system. However, the structure of DEQ helps us to circumvent this limitation. The sequence of the sampling chain can simultaneously converge to fixed point estimates to produce images $x_t$ at different time steps $t$. As seen in Figure 1 of the main paper, we indeed get low values of $\||h(x_{0:T}) - x_{0:T}\||_2$ which is indicative of good convergence. Further, Figure 2 visualizes the states of DEQ at the equilibria.
>
> > _I understand that in the DDIM and this paper, the authors use $\alpha$ to denote $\overline{\alpha}$ in the DDPM paper. However, the authors are suggested to explicitly point it out to avoid possible confusions of readers._
>
> We apologize for the confusion in notation between  $\alpha$ to denote $\overline{\alpha}$. We have updated the latest draft to reflect this. (Line 89-90)
>
> Thank you for pointing out our typos and grammatical errors. We have updated the draft to reflect your suggestions. We are happy to answer any follow up questions.

---

> > ### Comment · Reviewer_MgS2 · 2022-08-08
> > **Thanks for the response**
> >
> > Sorry that I misunderstood the DEQ formulation.  Would raise my rating accordingly.

---

### Meta-Review · Area_Chair_BFcE · 2022-08-27

**Recommendation:** Accept
**Confidence:** Certain

**Metareview:**

This paper develops a variant (deep equilibrium model (DEQ)) of an existing method (DDIM) with the goal of efficient sampling and model inversion while still maintaining the performance. In particular, it treats the sampling step sequence of DDIM（denoising diffusion implicit model, where the sampling step is deterministic and no longer a Markov chain.) as a fixed-point set, so that a fixed-point solver (e.g., Anderson acceleration, etc.) is able to jointly minimize the entire sampling chain.

The committee all agree that the methodology proposed in this work, although is built on prior work, is novel. The presentation of the paper is clear and the reported results are promising.  The committee appreciates the authors' effort in both revising the manuscript and providing a conclusive response. Therefore, we recommend acceptance of this manuscript.



**Award:**

No

---

### Decision · Program_Chairs · 2022-09-14

Accept